# The stabilized Pol31–Pol3 interface counteracts Pol32 ablation with differential effects on repair

Kenji Shimada, Monika Tsai-Pflugfelder, Niloofar Davoodi Vijeh Motlagh, Neda Delgoshaie, Jeannette Fuchs, Heinz Gut, Susan M Gasser

DNA polymerase δ, which contains the catalytic subunit, Pol3, Pol31, and Pol32, contributes both to DNA replication and repair. The deletion of *pol31* is lethal, and compromising the Pol3–Pol31 interaction domains confers hypersensitivity to cold, hydroxyurea (HU), and methyl methanesulfonate, phenocopying *pol32Δ*. We have identified alanine-substitutions in *pol31* that suppress these deficiencies in *pol32Δ* cells. We characterize two mutants, *pol31-T415A* and *pol31-W417A*, which map to a solvent-exposed loop that mediates Pol31–Pol3 and Pol31–Rev3 interactions. The *pol31-T415A* substitution compromises binding to the Pol3 CysB domain, whereas Pol31-W417A improves it. Importantly, loss of Pol32, such as *pol31-T415A*, leads to reduced Pol3 and Pol31 protein levels, which are restored by *pol31*-W417A. The mutations have differential effects on recovery from acute HU, break-induced replication and trans-lesion synthesis repair pathways. Unlike trans-lesion synthesis and growth on HU, the loss of break-induced replication in *pol32Δ* cells is not restored by *pol31*-W417A, highlighting pathway-specific roles for Pol32 in fork-related repair. Intriguingly, CHIP analyses of replication forks on HU showed that *pol32Δ* and *pol31-T415A* indirectly destabilize DNA pol α and pol ε at stalled forks.

## Introduction

DNA Polymerase δ (Pol δ) cooperates with DNA Pol α to synthesize the lagging strand during genomic replication, yet it also plays essential roles in leading strand synthesis and in many DNA repair pathways (Zhou et al, 2019) (reviewed in Prindle and Loeb [2012] and Nicolas et al [2016]). In budding yeast, as in most eukaryotes, DNA Pol δ contains a large catalytic subunit, Pol3, and two regulatory subunits, Pol31 and Pol32 (in humans, POLD1, POLD2, and POLD3, respectively) (Gerik et al, 1998). Mammalian DNA Pol δ also harbours a smaller subunit, POLD4, that appears to serve as an inhibitor of Pol δ processivity (Zhang et al, 2007, 2013). In budding yeast, *POL3*

and *POL31* genes are essential, whereas *POL32* is nonessential (Gerik et al, 1998). However, *pol32*-deficient strains are hypersensitive to impediments of replication fork progression, provoked by alkylating agents (methyl methanesulfonate or MMS) or hydroxyurea (HU), and are compromised for break-induced replication (BIR), a recombination event that allows restart at broken replication forks (Vijeh Motlagh et al, 2006; Hanna et al, 2007; Lydeard et al, 2007). In fission yeast, the Pol32 homologue is encoded by the essential gene *cdc27*, and like mammalian Pol δ, the complex contains a fourth subunit (p12 or Cdm1) that is nonessential for growth (Reynolds et al, 1998).

The exact roles of the second and third Pol δ subunits, Pol31 and Pol32, in holoenzyme function remain unresolved. A purely structural function, that is, that of linking the catalytic Pol3 to the third subunit Pol32, has been proposed for Pol31. That function alone, however, is inconsistent with the observation that *POL31* is an essential gene in yeast, whereas *POL32* is not. Here, we characterize the functions of the Pol δ second subunit, Pol31, at replication forks under DNA damage stress, its ability to compensate for the loss of the Pol32 subunit, and their roles in the survival of replication stress.

The regulation of DNA Pol δ is of central interest to cancer biology, as *POLD2* and *POLD3* are commonly amplified in human tumours (Beroukhim et al, 2010), and both germline and sporadic mutations were found in *POLD1* (reviewed in Nicolas et al [2016] and Fuchs et al [2021]). The *POLD1* mutations are associated with defective proofreading activity and result in a hypermutator phenotype (Palles et al, 2013; Rayner et al, 2016; Zhang et al, 2020). Such mutations have been linked to increased tumor incidence and decreased survival in mouse models (Goldsby et al, 2001). The physiological impact of POLD2 and POLD3 overexpression is unclear (reviewed in Nicolas et al [2016] and Fuchs et al [2021]).

It is of particular interest to understand the role of DNA Pol δ in the survival of replication stress, as most oncogene transformed human cells harbor signs of persistent replication stress, detected as S-phase checkpoint kinase activation (Bartkova et al, 2005; Macheret & Halazonetis, 2015) The checkpoint activation arises from an inappropriate origin firing and a failure to coordinate

---

Friedrich Miescher Institute for Biomedical Research, Basel, Switzerland

Correspondence: Susan.gasser@fmi.ch
Jeannette Fuchs's present address is Debiopharm, Lausanne, Switzerland
Susan M Gasser's present address is Fondation ISREC, Lausanne, Switzerland

replication with transcription (Gorgoulis et al, 2005, Macheret & Halazonetis, 2018). Consistent with genetic studies in yeast and flies, which indicate an important role for Pol32 in the maintenance of genome stability during replicative stress (Huang et al, 2002; Hanna et al, 2007; Lydeard et al, 2007; Ji et al, 2019), a study in human U2OS cells showed that POLD3 and POLD4 are required for processive DNA synthesis in cells transformed by cyclin E overexpression (Costantino et al, 2014). However, they also showed that POLD3 drives a significant fraction of genomic copy-number alternations generated under prolonged conditions of oncogene-induced replication stress. The authors propose that BIR mediated by POLD3 at compromised replication forks is responsible for genomic duplications in human cancers (Costantino et al, 2014). It remains unclear whether the role of Pol δ during replication stress is restricted to BIR-mediated restart, as Pol δ has also been implicated in translesion synthesis (TLS) and a range of other repair pathways (Giot et al, 1997; Hirota et al, 2016). Moreover, downregulation of POLD1 and POLD3 do not produce the same profiles of genome instability in cultured human cells: shPOLD3 generates R-loop associated DNA damage, whereas shPOLD1 does not (Tumini et al, 2016).

Genome-wide epistasis studies have shown that even though the deletion of POL32 in budding yeast is tolerated in the absence of DNA damaging agents, it is synthetic lethal with many factors involved in replication checkpoint response, namely, the reduction in the Mec1-Ddc2 checkpoint kinase activity, Mrc1, Mre11, and the RecQ helicase Sgs1, especially on HU (Tong et al, 2001) reviewed in Hustedt et al [2013]. Moreover, pol32Δ shows a strong conditional lethality on HU with these factors, and with the loss of 9–1–1 complex (Rad17, Ddc1, and Mec3) and its loader, Rad24, both of which contribute to the activation of the Mec1–Rad53 checkpoint cascade (Hustedt et al, 2015). Finally, pol32Δ was shown to be synthetically lethal on 0.1 M HU with a number of chromatin factors required for double-strand break (DSB) repair in the S phase, as well as the E2 Ubiquitin ligase Ubc13 (Hanna et al, 2007; Karras & Jentsch, 2010). Unfortunately, because POL31 deletion is lethal, similar EMAP studies could not be used to study synergistic lethality with the second Pol δ subunit.

To probe the function of Pol31, an alanine-scanning mutagenesis of conserved amino acids within Pol31's most conserved subdomains was carried out (Vijeh Motlagh et al, 2006). Although 45 of these alanine substitutions did not confer overt phenotypes, six novel temperature-sensitive (ts) or cold-sensitive (cs) pol31 alleles were isolated, which mapped to the conserved regions III, IV, VII, VIII, or IX in the linear gene structure (Reynolds & MacNeill, 1999; Vijeh Motlagh et al, 2006). The deletion of genes encoding fork stabilizing factors, namely, SGS1, RAD52, SRS2, MRC1, or RAD24, was deleterious in combination with the ts pol31 alleles, but not with those that were temperature-permissive (Vijeh Motlagh et al, 2006). This argued that both recombination and checkpoint functions are required for processing the DNA lesions or structures that arise because of defects in DNA Pol δ. Intriguingly, all ts-/cs- pol31 alleles, regardless of their phenotype as a single mutant, became dependent on the presence of an intact Pol32 subunit, suggesting partial complementarity between the two smaller subunits of the Pol δ complex (Vijeh Motlagh et al, 2006).

In addition to the ts- and cs-alleles, three temperature-permissive pol31 alleles (alanine substitutions at D297, W417, and both E463 and F464, hereafter pol31-D297, pol31-W417, and pol31-E463F464) were identified as suppressors of the cold-, HU-, and MMS-sensitivities of pol32Δ cells. This suggests a mutation-dependent gain of function that bypasses important functions normally provided by Pol32. None of the mutations that suppress pol32Δ mapped to the interface of Pol31 with Pol32, nor to the putative (PCNA) proliferating cell nuclear antigen-binding interface. Rather, we find that pol31-W417 enhances the interaction between Pol31 and the catalytic subunit Pol3, whereas an adjacent cs-allele, pol31-T415, compromises this same interaction. Structural modeling shows that these two amino acids identify contacts between Pol31 and the C-terminal CysB Fe-S cluster of Pol3. The same mutations also affect interaction with the related C-terminal domain (CTD) of the translesion polymerase (TLS) Rev3 (pol ζ [Sanchez Garcia et al, 2004; Makarova et al, 2012]).

We find that enhancement of Pol31–Pol3 interaction correlates both with resistance of MMS, and with improved abilities to maintain replication fork stability and resume replication after a prolonged exposure to HU. On the other hand, it does not restore BIR significantly in a pol32 deletion strain. We find that all three replicative polymerases are destabilized at stalled replication forks in pol32Δ on HU, and that stable binding of DNA pol ε is partially suppressed by the pol31-W417 allele. The hypersensitivities of pol32Δ correlate with reduced levels of Pol3 and Pol31 provoked by loss of Pol32, and this is compensated by pol31-W417. Thus, we propose that a critical threshold of DNA Pol δ is required during replication stress, and that the stability of Pol3–Pol31 interaction reduces protein loss. Intriguingly, re-establishment of Pol3 level does not compensate for pol32Δ in BIR. Our observations are of particular interest with respect to human cancers, which commonly elevate POLD2 and POLD3 levels, perhaps to compensate elevated levels of replication stress (Fuchs et al, 2021).

## Results

Based on protein sequence comparisons of the POLD2/Pol31 subunit across eukaryotes, 10 conserved regions (I-X; Fig 1A [Reynolds & MacNeill, 1999]) were identified and the amino acids with the highest conservation were mutated one by one to alanine (Vijeh Motlagh et al, 2006). Three alanine substitution mutants in Pol31, namely, D297, W417 and the double mutant E463F464, which map to the domains VI, VIII, and X, respectively, showed no enhanced sensitivity to either HU, MMS, nor to growth at low temperature, unlike pol32Δ, which is highly sensitive to all three conditions (Fig 1A–C). However, when combined with pol32Δ, each of these alleles was able to suppress the drug and cold sensitivity of pol32Δ (Figs 1B and S1).

Based on the crystal structure of the human orthologue of Pol31-Pol32N-term (Baranovskiy et al, 2008), we built a homology model of Pol31, on which we indicate the positions of the novel pol31 mutations (Fig 1C). The three suppressor alleles do not cluster spatially, although D297 and E463F464 are not far from the Pol32 binding surface in the folded structure (Fig 1C). In contrast, the third suppressor mutation (pol31-W417) mapped to a solvent exposed loop of Pol31, which has the highest conservation score within Pol31. Intriguingly, two amino acids away, we identified a T415A substitution that conferred strong HU-, MMS-, and cold-sensitivity in an otherwise wild-type background (Fig 1B). Thus, two mutations with nearly opposite effects map to one exposed loop: one suppressing pol32Δ phenotypes (W417A) and one (T415A) mimicking them.

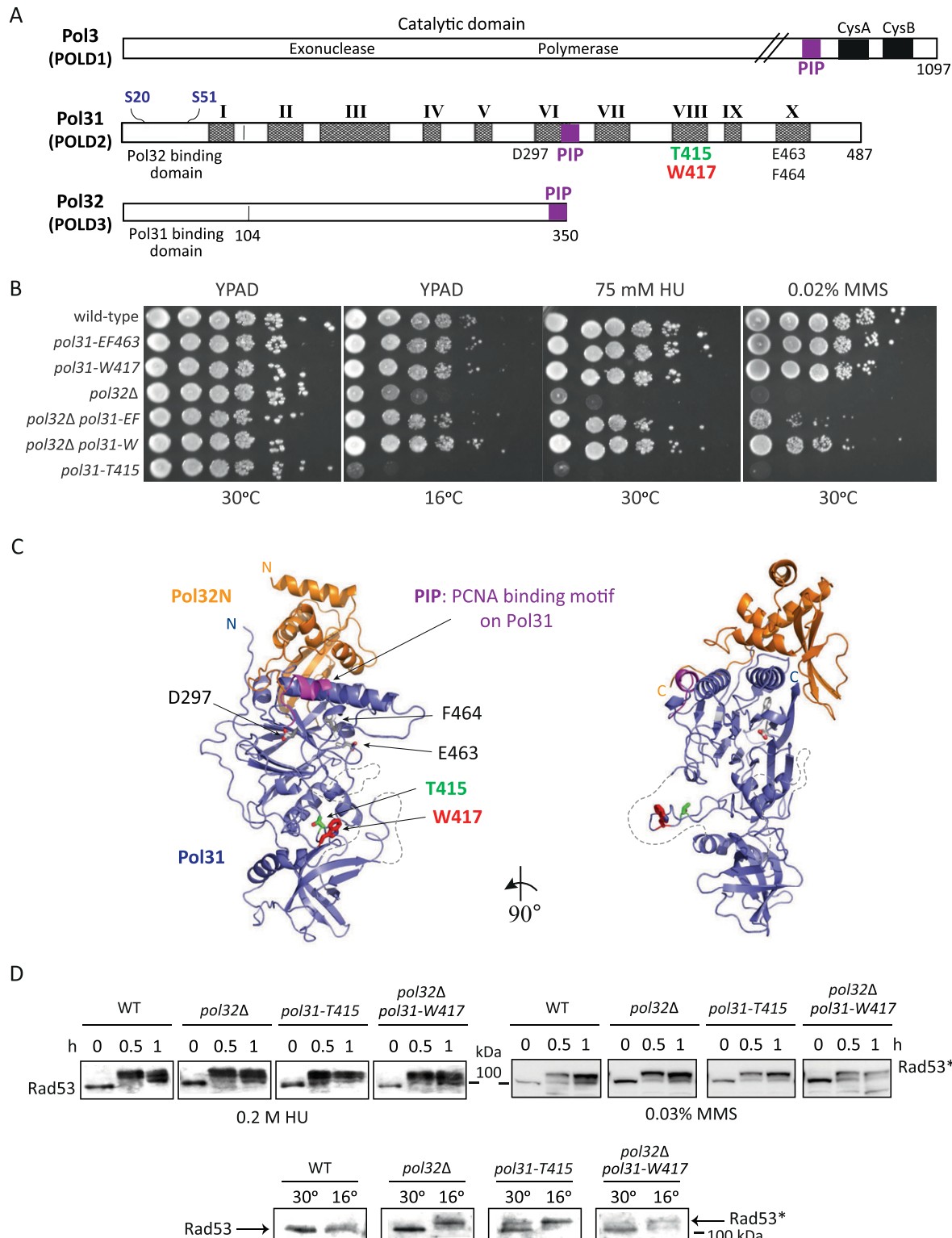

**Figure 1. Identification and position of *pol31* mutants that suppress the HU-, MMS-, and cold sensitivity of *pol32Δ* and the *pol31-T415* mutant which exhibits HU-, MMS-, and cold sensitivity.**
**(A)** Linear map of the three subunits of DNA Pol δ: Pol3, Pol31, and Pol32, with interacting domains, CysA and CysB Zn or Fe-S fingers and the conserved domains of Pol31 (I-X, shaded). Mutations relevant to this study are indicated. The E463 and F464 mutations are coupled. S20 and S51 are phosphoacceptor sites for Mec1 kinase in response to HU. PIP: PCNA interaction sites. **(B)** Serial dilution (10× series) drop assays demonstrate the sensitivities of the *pol31-T415* and *pol32Δ* mutants, and the ability of *pol31-W417*, and *pol31-E463F464* mutants to suppress the lethality of *pol32Δ* on 75 mM HU or 0.02% MMS or at 16°C. Strains used are GA-4629, –9784, –9782, –4761, –4639,

We can exclude the possibility that damage sensitivity or suppression is linked to altered checkpoint activation because Rad53 (CHK2) phosphorylation after HU treatment is intact in *pol32Δ* as well as in *pol31-T415*, and in the *pol31-W417 pol32Δ* double mutant (Fig 1D). Intriguingly, we note that growth at 16°C alone, in the absence of external agents, can trigger checkpoint activation in both *pol32Δ* and the *pol31-T415* mutants, although the exact mechanism is unclear. There is a slight drop in activated Rad53 level at 1 h on 0.2 M HU and at 16°C (small arrow) in the *pol31-W417 pol32Δ* strain that correlates with improved survival, and possibly with more rapid repair (Fig 1B and D).

### The pol31-W417 improves and pol31-T415 compromises Pol3–Pol31 binding in the absence of Pol32

The solvent-exposed loop of Pol31 containing both T415 and W417 is found at the interface of Pol3 with Pol31 in the recently published crystal structures of Pol δ (Jain et al, 2019) (see also Lancey et al [2020]). We therefore examined the impact of these mutations on Pol31 protein interaction with its most likely interaction partner, the Pol3 CTD (Sanchez Garcia et al, 2004), by yeast two-hybrid and pull-down assays. The CTD of Pol3 contains two metal-chelating 4-Cys motifs called CysA and CysB (Netz et al, 2011). CysA is a Zn-finger involved in PCNA binding (Acharya et al, 2011; Khandagale et al, 2020), whereas the CysB is an Fe-S center (Jain et al, 2019) that can bind Pol31 (Sanchez Garcia et al, 2004). Quantitative two-hybrid assays using CysA and CysB domains of Pol3 and full length Pol31 show that the *pol31-T415* mutation compromises the interaction of Pol31 with the Pol3-CTD, whereas the *pol31-W417* mutation does not (Fig 2A). Moreover, when challenged with only the CysB domain of Pol3, we find that the *pol31-T415* mutation compromises the interaction of Pol31 with the Pol3–CysB, whereas the *pol31-W417* mutation actually increases Pol31–Pol3–CysB binding (Fig 2B). This was true both in the presence and in the absence of Pol32, confirming that the effects of the mutations are direct, and not achieved by trimer formation. We rule out that this is due to differential stability of the LexA–Pol31 mutant proteins by Western blot (Fig S2A). We note that the levels of two-hybrid interaction are reduced by *pol32Δ*, when the smaller CysB prey is used (Fig 2B), probably because CysA contributes to the binding of PCNA (Acharya et al, 2011; Khandagale et al, 2020), which in the presence of Pol32 provides an additional bridge to Pol31. The direct interaction of Pol31 and Pol3 in the absence of Pol32, that we monitor here, is in contrast to the situation in *Schizosaccharomyces pombe*, in which the Pol32 homologue, Cdc27, is needed for interaction of the other two subunits (Sanchez Garcia et al, 2004).

Pol31 has been implicated in tethering the holo-Pol δ to PCNA (Acharya et al, 2011), thus we checked the impact of the Pol31 mutations on PCNA interaction. Again there is a drop in Pol31-PCNA binding with pol31-T415, which is not found for pol31-W417 (Fig 2C). Unlike the interaction of Pol31 with Pol3, we find that the interaction between Pol31 and PCNA requires the presence of endogenous Pol32 (Fig 2C). This is not an artefact of the *pol32Δ* background since

the Pol32 N terminus (aa 1–350) binds all three forms of Pol31 in the *pol32Δ* strain equally (Fig 2C), as predicted by previous studies (Johansson et al, 2004). Given that Pol31–PCNA interaction requires Pol32, a drop in PCNA interaction cannot explain the suppression of *pol32Δ* phenotypes by *pol31-W417*. Rather, we propose that these mutations in the Pol31 loop (T415 and W417) reflect the reduction and enhancement, respectively, of Pol31–Pol3 interactions, and thus only indirectly affect PCNA binding.

Given the high degree of homology between the CTD of Rev3 and Pol3, we next tested the effects of these mutations on the interaction of Pol31 and the CTD of the TLS polymerase, Rev3 (Pol ζ) (Sanchez Garcia et al, 2004). We confirm that Pol31 binds the CTD of Rev3, and we see similar fluctuations in the interaction of Pol31 with Rev3-CTD in the presence of the T415 and W417 mutations (Fig 2D). The significance of this interaction for the survival on MMS is examined below.

We confirmed the two-hybrid results with Pol3 with in vitro pull-down assays using whole yeast cell extracts from cells expressing epitope-tagged wild-type or mutant forms of Pol31, which were probed with bacterially expressed MBP-Pol3-CTD bound to Amylose beads (see the Materials and Methods section; Fig S2B). Although the differential interaction was not observed in pull-down assays using Pol31 expressed in wild-type cells grown at 30°C (Fig S2C), when yeast cells were grown at 16°C before extraction of Pol31, we monitored a significantly reduced pull-down of HA-pol31-T415 by the Pol3-CTD, whereas HA-pol31-W417 and wild-type HA-Pol31 bound Pol3-CTD with equal efficiency (Fig 2E). We rule out that it arises from differential expression or stability of the Pol31 variants in the whole cell extracts by Western blot (Fig S2E and F). The reduced affinity for pol31-T415 for Pol3 is much more pronounced in the absence of Pol32, suggesting that in its presence (i.e., in wild-type extracts), a trimeric Pol3–Pol31–Pol32 complex stabilizes the interaction.

It is unclear why growth at 16°C reveals the reduced pol31–T415–Pol3 interaction since the actual pull-down assay is carried out at 4°C in all cases. Nonetheless, this reduced interaction correlates with the cs of the *pol32Δ* and *pol31-T415* mutant cells (Fig 1B). In conclusion, both two-hybrid and pull-down assays showed that the Pol3–Pol31 interface is compromised in the pol31-T415, but not pol31-W417 protein, and that the defect is particularly pronounced in the absence of Pol32 (Fig 2E).

### pol31-W417 can suppress the MMS sensitivity of pol32Δ through post-replicative repair (PRR) without TLS polymerase, Rev3

We hypothesized that the altered binding of Pol31 to the related Fe-S clusters in the CTDs of Pol3 and Rev3, the TLS polymerase, explains the sensitivity of *pol31-T415* to MMS and HU, as well as the *pol31-W417*–mediated suppression of *pol32Δ* hypersensitivity at 16°C, or on HU and MMS. In recovery assays, in which the cells synchronized in G1 are released into MMS for varying times before plating on drug-free YPAD, we observed strongly impaired cell survival for the cs mutant

−4636, and −4630. See Fig S1 for similar results with *pol31-D297*, and its comparison with the other suppressor alleles. **(C)** Homology model of the Pol31–Pol32N complex. Pol31 (blue) and Pol32N (orange) are displayed as cartoon models in two orientations rotated around a vertical axis by 90°. Specific residues which were mutated in this study are shown as sticks in atom colors (T415, green; W417, red; D297, and E463-F464, grey). The Pol31 PCNA binding motif is highlighted (magenta) and the N- and C termini are labeled. Flexible loops which were not included in the homology modeling calculation are shown as dashed grey lines. **(D)** Rad53 phosphorylation upshift (Rad53*) monitors checkpoint kinase activation after treatment with HU or MMS at 30°C, or incubation at 16°C, which causes lethality in *pol32Δ* as well as *pol31-T415* mutants. Checkpoint activation is intact in *pol32Δ*, *pol31-T415*, and *pol32Δ*, *pol31-W417* double-mutants after treatment with HU, MMS, or during incubation at 16°C as seen by Rad53 upshift. Rad53 upshift in *pol32Δ* mutant at low temperature is partially suppressed by additional mutation of *pol31*-W417. **(B)** Strains as in (B).

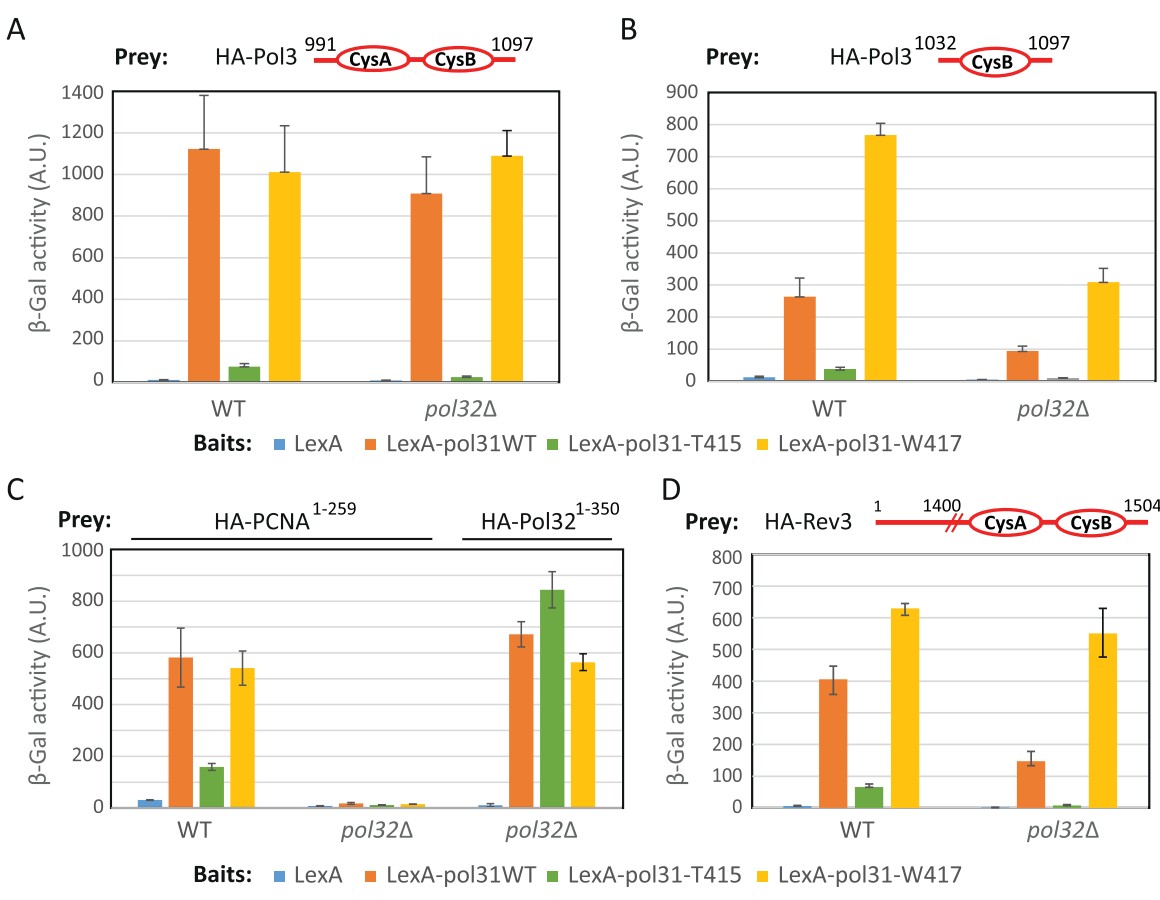

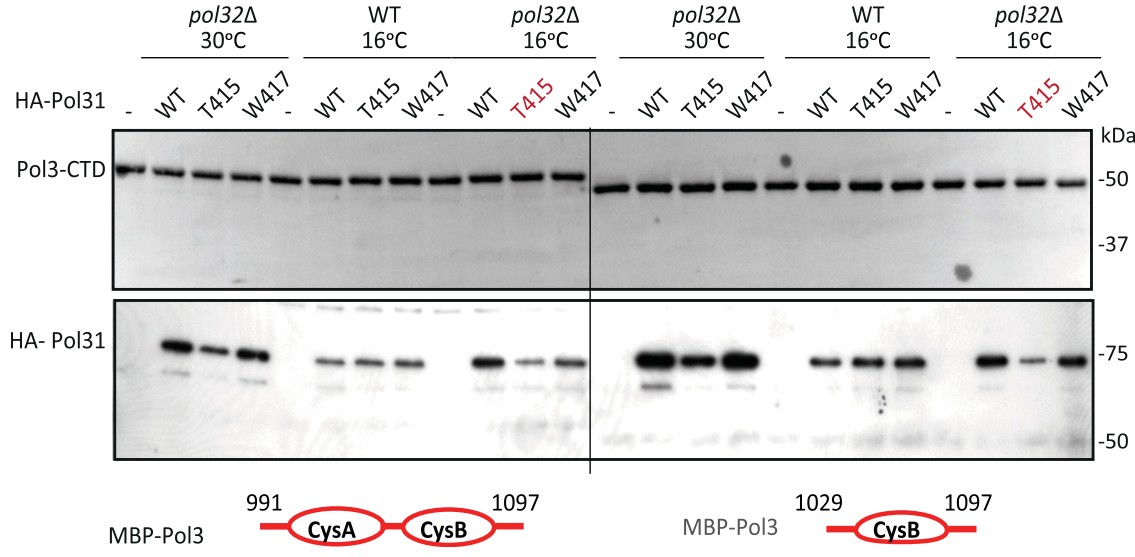

E    **In vitro pull-down of HA-tagged Pol31 variants with purified Pol3-CTD**

**Figure 2.   Pol31 interaction with the C-terminal domain (CTD) of DNA Pol3 is sensitive to W417 and T415 mutations.**
**(A)** Yeast two-hybrid interaction performed by standard methods is quantified by *β*-galactosidase units in solution (A.U.) which correlate linearly with the strength of bait and prey interaction. The assay is performed in the indicated backgrounds (wild-type or *pol32Δ*) with the indicated bait and prey constructs. Wild-type Pol31 and *pol31*-W417, but not *pol31*-T415, bind the Pol3 C terminus (991-1097) bearing the paired CysA and CysB domains, with or without endogenous Pol32. **(A, B)** As (A), but the yeast two-hybrid prey contains only the more distal CysB Fe-S cluster of the Pol3 C terminus (1032-1097). The interaction is again compromised by the *pol31-T415* mutation. Although the interaction between Pol3-CysB domain and Pol31 is slightly weakened in the absence of Pol32, *pol31-W417* mutation increases its affinity, irrespective of

*pol31-T415*, as was observed for *pol32Δ* cells (Fig 3A). Moreover, *pol31-W417* suppressed the MMS sensitivity of *pol32Δ* over 10-fold. There are two pathways used by yeast undergoing replication to repair alkylated bases after the passage of the replicative polymerases, defining PRR pathways. The first is TLS which involves the recruitment of error-prone TLS polymerases such as Pol ζ (Rev3), whereas the second is largely error-free, and involves bypassing the mutation by copying the intact strand, through an exchange that requires Rad5, Mms2, and Ubc13 (Fig 3B). Key to pathway choice is the ubiquitination status of PCNA (Hoege et al, 2002). TLS requires the covalent mono-ubiquitination of K164 of PCNA which is catalyzed by the Rad6/Rad18 ubiquitin ligase complex Hoege et al, 2002; Gangavarapu et al, 2006; Carlile et al, 2009; Xu et al, 2015; Halas et al, 2016. The poly-ubiquitination of this K164 acceptor residue on PCNA, on the other hand, shunts the lesion to the error-free bypass pathway, triggered by Mms2, Rad5, and Ubc13. PCNA poly-ubiquitination activates a recombination-like DNA damage-avoidance mechanism (Fig 3B).

To see which branch of PRR is involved in the *pol31-W417* suppression of *pol32Δ*'s MMS sensitivity, we made double and triple mutants with key regulators of TLS and Error–error free lesion bypass. If the *pol31-W417* mutation suppresses by favoring TLS over the bypass pathway, we expect suppression to be lost in the *pol31-W417 pol32Δ* background when coupled with pathway-specific mutants. Starting with the upstream regulator, Rad18, we see that the ability of *pol31-W417* to suppress the sensitivity of *pol32Δ* to 0.01% MMS is indeed lost in the absence of Rad18 (Fig 3C). It is also unable to suppress *pol32* deletion in the absence of Rad5 and Mms2 (Fig 3D and E). In contrast, the loss of Rev3 did not compromise the ability *pol31-W417* to suppress the sensitivity of *pol32Δ* to MMS (Fig 3F), suggesting that increased binding of the TLS polymerase is not needed for the suppression phenotype. We confirmed the fact that *pol31-W417* can suppress the sensitivity of *pol32Δ* to MMS without recruiting Rev3, with a more quantitative colony outgrowth assay (Fig S3). Wild-type and mutant cells were synchronized in G1, released into 0.03% MMS for increasing amounts of time, and then plated on drug-free media to monitor colony forming units (CFUs) (c.f., Fig 3A). In this acute exposure and recovery assay, *pol31-W417* again suppressed *pol32Δ* sensitivity in the absence of Rev3 (Fig S3), arguing that the *pol31-W417* suppression pathway exploits an enhanced affinity of Pol31 for Pol3, stimulating either Error-free lesion bypass or TLS mediated by Pol δ, but not by Rev3. We note that in DT40 cells, DNA Pol δ has been implicated as an alternative polymerase for TLS in DT40 cells (Hirota et al, 2016).

### *pol31-W417* can suppress the HU sensitivity of *pol32Δ* independently of PRR regulation

Yeast cells bearing the *pol32Δ* or *pol31-T415* mutations are not only sensitive to MMS, but also to HU, which depletes deoxyribonucleotide triphosphates (dNTPs) and destabilizes Fe-S cluster proteins

(Huang et al, 2016). In addition to the serial dilution drop assay (Fig 1B), we monitored the ability of cells to recover from acute fork arrest triggered by exposure to 0.2 M HU, that is, to form colonies on HU-free media after a time-limited incubation with high levels of HU (Fig 4A). The recovery assay showed that *pol31-T415* strains are even more sensitive to HU than *pol32Δ* cells, whereas *pol31-W417* partially suppresses the *pol32Δ* defect (Fig 4A). Not surprisingly, the degree of sensitivity of *pol32Δ* and its suppression by *pol31-W417* are even more pronounced when cells are exposed to both MMS and HU (Fig 4B). Earlier work showed that DSBs arise from the combined treatment, which is not the case for either agent alone (Nagai et al, 2008). This may indicate that strand-invasion and fork restart are more strongly Pol32-dependent than TLS or the survival of fork stalling.

Again serial dilution drop assays were used to identify the pathways implicated in the suppression of *pol32Δ* deficiency on HU, for they allowed us to compare the growth patterns of single, double and triple mutants. Interestingly Pol32 is more sensitive to 25 mM HU than 20 mM (Fig 4C–F), and although the genes involved in PRR are themselves not sensitive to HU, we observe some additivity when *pol32Δ* is combined with *rad18Δ*, *rad5Δ*, or *mms2Δ*, but not with *rev3Δ*. Once again, the *pol32Δ*-linked sensitivity to HU was suppressed by *pol31-W417*, and this suppression was entirely independent of Rev3, Rad5, and Mms2, and largely independent of Rad18 (Fig 4C–F). This leads to a unique pathway of Pol δ-dependent fork survival on HU that is independent of TLS and error-free lesion bypass. In other words, the pathway by which Pol δ mediates survival of HU-induced stress is distinct from the role it plays in the repair of MMS-induced alkylation. Nonetheless, the same gain-of-function mutation, *pol31-W417*, is able to suppress the sensitivity.

### Pol3 and Pol31 levels drop in *pol32Δ* and are restored by *pol31-W417*

Studies in other organisms have shown a drop in steady-state levels of the catalytic and second subunits of Pol δ upon loss or down-regulation of the Pol32/POLD3 subunit (Murga et al, 2016; Tumini et al, 2016; Conde et al, 2019; Ji et al, 2019). We could show, however, in quantitative Western blots of total cell extracts from wild-type and *pol32Δ* cells synchronized in the S phase that the steady state levels of DNA Pol1 and Pol2 were nearly fourfold elevated in cells lacking Pol32, whereas the levels of Pol31 were reduced (Fig 5A). Whereas *pol32Δ* cells tend to accumulate in the late S phase in unsynchronized populations, in this study, we synchronized the cells by pheromone arrest and release for 30 min, minimizing cell cycle differences between wild-type and *pol32Δ* cells.

To see if Pol3 levels are indeed compromised by *pol32Δ* and/or *pol31-T415*, we tried to tag the catalytic subunit Pol3 with either C-terminal 9-Myc or 3-FLAG epitope tags, which have been used

Pol32. **(C)** Yeast two-hybrid assays monitoring Pol31 wild-type and mutant interaction with PCNA in a manner dependent on Pol32. *pol31-T415* reduces interactions with PCNA, but *pol31-W417* does not. Pol31 binds the Pol32 N terminus, independently of endogenous Pol32 and *pol31* mutations. **(C, D)** As (C), Yeast two-hybrid assays performed as above with the indicated baits and prey. This assay shows that the *pol31-T415* mutation strongly reduces affinity to the Rev3 CTD (the catalytic subunit of Pol ζ), whereas *pol31-W417* enhances it, independent of endogenous Pol32. Again *pol32Δ* partially weakens wild-type Pol31 and Rev3 interaction. **(E)** Pull-down assays for Pol31 and variants expressed in yeast extracts by purified MBP-Pol3 CTDs. The scheme for the pull-down assay is in Fig S2E. The cs mutant *pol31-T415* shows reduced binding to Pol3-991-1097 and with Pol3-1032-1097 than wild-type Pol31. The reduced binding is detected only in cell extracts lacking Pol32 (*pol32Δ*) irrespective of growth temperature. -, vector only; WT, expression of full-length HA-tagged Pol31; T415, expression of full length HA-tagged Pol31-T415 mutant; W417, expression of full length HA-tagged Pol31-W417 (HA-tagged Pol31 is 78 kD).

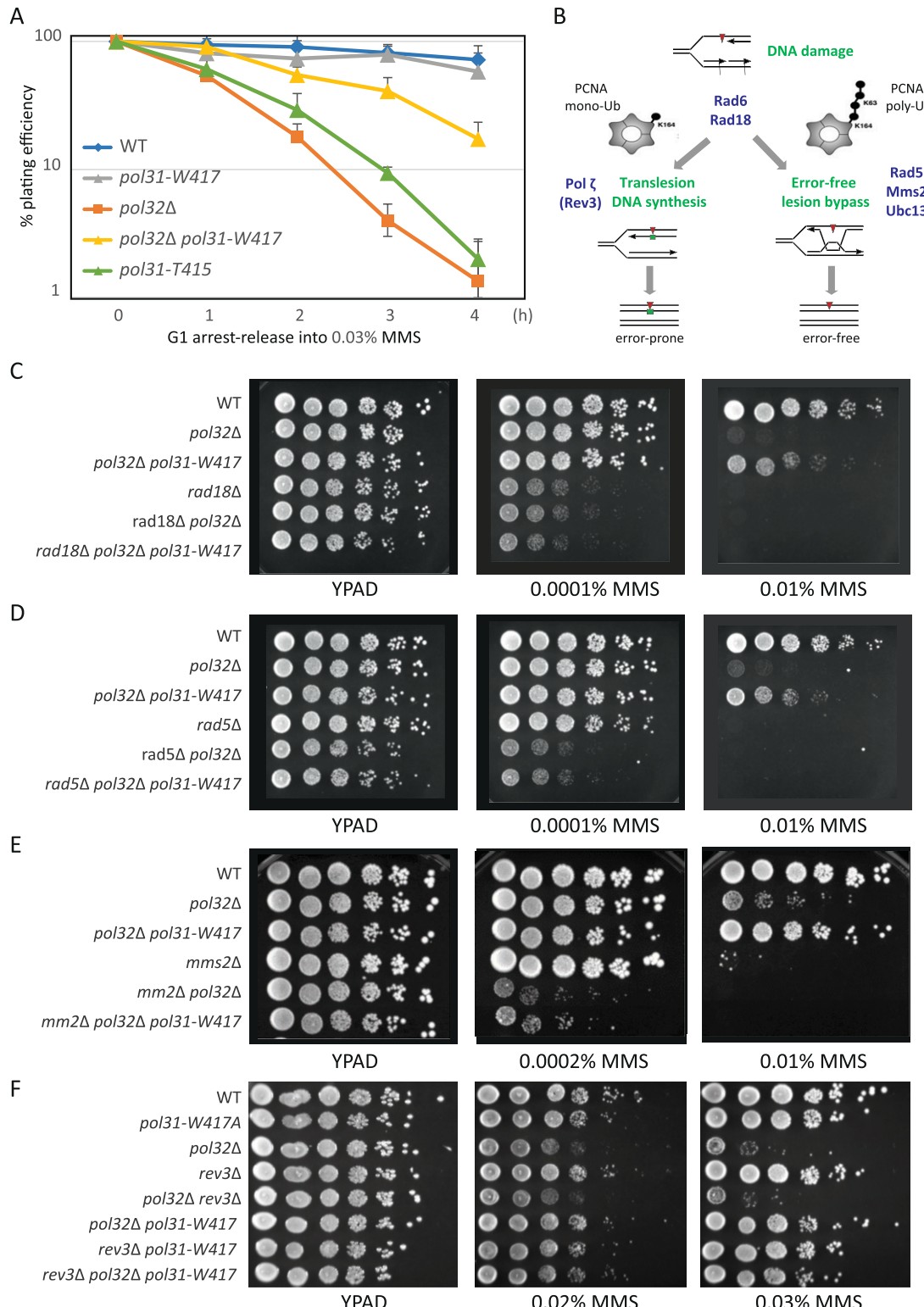

**Figure 3. Survival on MMS requires Pol32 and *pol31-W417* suppression acts through the Error-free lesion bypass pathway.**
**(A)** Quantitative recovery assay showing that the *pol31-W417* mutation suppresses the growth defects of *pol32Δ* in liquid culture on MMS. Cells were G1-arrested by α-factor, released into 0.03% MMS, sampled at indicated time-points and plated on YPAD plates. Colonies are scored after 3 d. **(B)** Post replication repair pathways and key regulatory proteins. **(C)** Serial dilution (10×) drop-assay showing *rad18Δ* combined with *pol31-W417* and *pol32Δ* mutants on YPAD plates with or without the indicated concentration of fresh MMS. *pol31-W417* cannot suppress the MMS hypersensitivity of *pol32Δ* in the absence of Rad18. **(C, D)** as (C), but for the *rad5Δ* mutant. The *rad5Δ* mutant is very sensitive to 0.01% MMS on its own and dead at 0.01% MMS. Although *pol31-W417* suppresses the MMS sensitivity of *pol32Δ* in the presence of Rad5, it cannot

successfully on other DNA polymerase genes. In a wild-type background, the Myc-tag and 3-FLAG tags enhanced the intrinsic sensitivity of yeast cells to MMS, and we were unable to combine the tagged Pol3 subunit with either the *pol31-T415* allele or with *pol32Δ* by mating, as the double mutants yielded dead spores (Fig S4A and B). Therefore, we used a CEN-plasmid based system to attach a small N-terminal Spot tag (peptide PDRVRAVSHWSS, ChromoTek) to the catalytic subunit of Pol δ, Pol3. The Spot tag was fused to an intact copy of *POL3* that was expressed from a single copy plasmid in both wild-type and *pol32Δ* cells (see the Materials and Methods section). The resulting Spot-Pol3 protein was recovered from wild-type and *pol32Δ* yeast cell lysates by binding to a vast excess of Spot-trap antibody on magnetic beads. Quantitation of the recovered Pol3 showed a drop in its steady state level upon *pol32* deletion (Fig 5A, right panel). All Westerns and protein extractions were highly reproducible, and the increase in DNA Pol1 and Pol2, as well as the reduction of DNA Pol3, was observed in repeatedly. We were also able to reproduce the observation that in cultured human Jurkat cells, shRNA against *POLD3* reduced the steady state levels of POLD1 and POLD2 (Fig 5B), reinforcing the observed drop in Pol3 and Pol31 upon *pol32* deletion.

To link the fluctuation of Pol3 level to the HU-, MMS-, and cold sensitivities of the *pol32Δ* mutant, we introduced the plasmid expressing *Spot-POL3* (pSpot-POL3) into *pol32Δ*, *pol31-T415* and the *pol31-W417 pol32Δ* double mutant strains. The tagged Pol3 protein level drops to 70% of its wild-type level in the *pol32Δ* background (Fig 5C), recapitulating the reduction observed in Fig 5A. Moreover, the Spot-Pol3 fusion level also dropped in the presence of the *pol31-T415* mutant, and the addition of *pol31-W417* to the *pol32Δ* mutant background restored wild-type Pol3 levels (Fig 5C). Thus, the destabilization and restoration of the level of the catalytic Pol3 subunit correlates with the phenotypes associated with *pol32Δ*, *pol3-T415*, and its suppression by *pol3-W417*.

To confirm this, we examined the physiological impact of having an extra copy of Pol3 provided by the CEN-plasmid borne *Spot-POL3* construct. In wild-type cells, the presence of p*Spot-POL3* had no effect on viability nor sensitivity to replication fork stress (Fig 5D, see Sc-Leu). However, having a second copy of Pol3 (p*Spot-POL3*) itself could suppress the sensitivity of the *pol32Δ* and *pol31-T415* strains to HU-, MMS-, and growth at 16°C (Fig 5D). As shown above, the presence of *pol31-W417* suppressed the *pol32Δ* sensitivities, in a manner that appears to be epistatic with the presence of p*Spot-POL3* (growth of the double mutant was identical with or without the plasmid; Fig 5D). We conclude that the loss of Pol32 destabilizes both Pol31 and the Pol3 catalytic subunit in yeast, as shown in human cells (Fig 5B). Moreover, both the reduced Pol3 levels found in *pol32Δ* and the ensuing replication stress sensitivity are suppressed either by the *pol31-W417* mutation or by adding an extra copy of functional Pol3. We conclude that the increased affinity between Pol31 and the Pol3-CTD found in *pol31-W417* stabilizes Pol3

under conditions of replication fork stress. The same can be achieved by doubling *POL3* copy number (adding one CEN-plasmid borne copy).

## DNA polymerase chromatin immunoprecipitation (ChIP) at stalled replication forks

It remained unclear what impact reduced levels of Pol3 might have on replication forks stalled by high HU concentration. We and others have previously published ChIP methods that monitor DNA Pol α and Pol ε levels quantitatively at replication forks following arrest on 0.2 M HU (Cobb et al, 2003, 2005; Lucca et al, 2004; Lou et al, 2008). Replication fork collapse or fork reversal correlates with the displacement of DNA polymerases near origins, thus intuitively, the stabilization of polymerases at forks stalled by low dNTP level should facilitate fork restart and cell survival. ChIP for DNA Pol δ has not been previously published; therefore, we examined whether Pol3 is also stabilized at the stalled replication fork on HU. Because no appropriate anti-Pol3 antibody is available, we tagged the endogenous *POL3* gene C-terminally with the 9-Myc epitope in an otherwise wild-type background. We then monitored quantitatively the proteins bound near an early firing origin (ARS607) by ChIP-qPCR (Fig 6A), in cells released into S phase with a high HU concentration. We detect a low but nonetheless reproducible enrichment of Pol3 at the stalled fork, but it did not migrate far along the DNA fiber (Fig 6B). This low enrichment is not surprising given the fact that C-terminally tagged Myc-Pol3 shows compromised growth on HU and MMS (Fig S4), precluding meaningful studies with this construct.

The modest enrichment of Pol3 by ChIP, and the fact that we could not generate endogenously tagged Pol3 in either the *pol32Δ* or *pol31-T415* backgrounds (Fig S4B), led us to test instead whether the reduced survival of acute HU arrest reflects a loss of DNA Pol α or Pol ε at the replication fork. We performed ChIP on G1-synchronized cells released into 0.2 M HU to monitor the presence of Pol1-3HA and Pol2-13Myc, the catalytic subunits of DNA Pol α and Pol ε. The abundance at 2, 4, 6 or 10 kb from ARS607 was normalized to the polymerase level at a site 14 kb away from the origin, which is taken as background, given that it is halfway to the next origin. We see that in wild-type strains, the polymerases are enriched 5- to 10-fold within 5 kb of the origin on HU, and stay bound for at least 1 h. In mutants such as *sgs1Δ*, *mrc1Δ*, or *mec1Δ*, the polymerases are significantly less stable (Cobb et al, 2003, 2005; Lou et al, 2008). Similar ChIP analyses show that in *pol32Δ*, DNA Pol α (Fig 6C) as well as DNA Pol ε (Fig 6D) lose their stable enrichment, even though the overall cellular levels of these catalytic subunits is elevated (Fig 5A). This argues that holo-Pol δ contributes to replisome stability as a whole at HU-stalled forks.

To see if this phenomenon is linked to alterations in the Pol31-Pol3 interface, we tested the *pol31-T415* mutant, and the *pol31-W417 pol32Δ* double mutant for Pol2-Myc recovery at the stalled fork. ChIP analyses show that, as for *pol32Δ*, Pol2 was destabilized at a stalled fork by

---

suppress MMS sensitivity in its absence (see triple mutant). **(E)** as C, but for the *mms2Δ* mutant. Whereas *mms2Δ* survives 0.0002% MMS, it is highly sensitive to 0.01% MMS. In combination with *pol32Δ* it shows hypersensitivity at both concentrations of MMS. Whereas there may be slight suppression by *pol31-W417* at 0.0002% MMS, at 0.01% MMS, the strong suppression of *pol32Δ* sensitivity is lost in the absence of Mms2. **(C, F)** as (C), with *rev3Δ* mutant. The *rev3Δ* mutant on its own is not sensitive to 0.02% or 0.03% MMS, but in combination with *pol32Δ* and shows no additivity in the double mutant. *pol31-W417* efficiently suppresses the growth defects of *pol32Δ* in the absence of Rev3 (triple mutant).

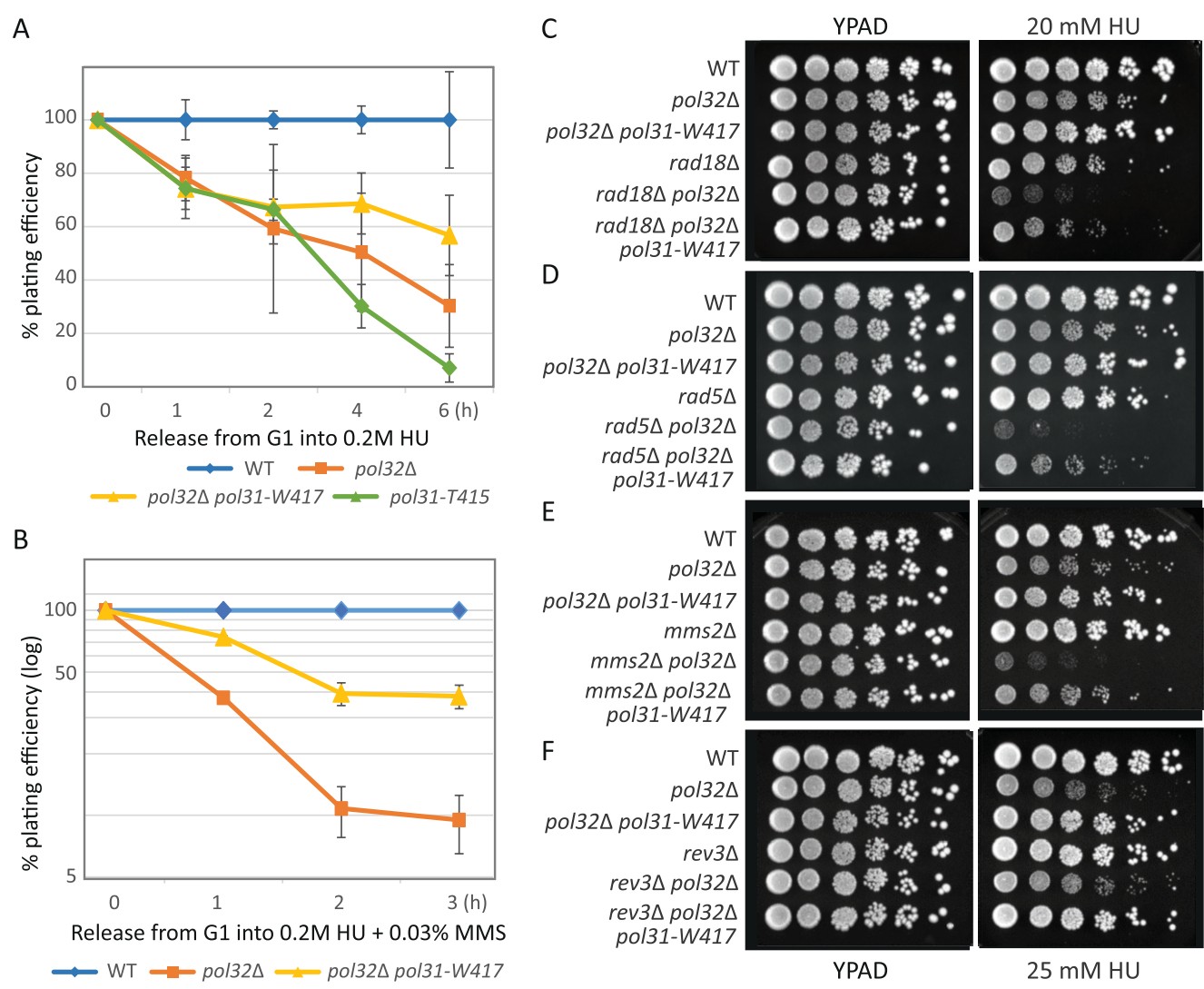

**Figure 4. Survival on HU: *pol31-W417* suppresses defects of *pol32Δ* independently of post replication repair components.**
**(A)** Colony outgrowth assay for exposure to acute levels of HU for the indicated times. Isogenic strains bearing the indicated mutations were synchronized in G1 and released into 0.2 M HU for 1–6 h before plating on YPAD. Colony scoring was carried out in triplicate after 3 d at 30°C. **(B)** As A, but cells were also treated with 0.03% MMS during the exposure to 0.2 M HU for 1, 2, or 3 h. After plating on reagent-free YPAD, colony scoring was carried out in triplicate after 3 d at 30°C. **(C)** Serial dilution (10×) drop-assay on YPAD with 20 mM HU showing *rad18Δ* combined with *pol31-W417A* and *pol32Δ* mutants. The *pol32Δ* and *rad18Δ* mutants on their own are slightly sensitive to 20 mM HU and when combined, *pol32Δ rad18Δ* shows synergistic sensitivity. The *pol31-W417A* allele suppresses lethality in combination with *pol32Δ* and in the triple mutant, *rad18Δ pol32Δ pol31-W417*, indicating that it does not require Rad18 for suppression of *pol32Δ*. **(C, D)** As (C) but with *rad5Δ* mutant. The *rad5Δ* mutant on its own is insensitive to 20 mM HU, yet is synergistically sensitive together with *pol32Δ*. *pol31-W417* suppresses the growth defects of *pol32Δ* on HU in the absence of Rad5 (see triple mutant). **(C, E)** As (C), but with *mms2Δ* mutant on 25 mM HU. The *mms2Δ* mutant has no sensitive to 25 mM HU, yet shows additivity when combined with *pol32Δ*. The *pol31-W417* allele efficiently suppresses the growth defects on HU in the absence of Mms2 (see triple mutant). **(E, F)** As (E), but with *rev3Δ* mutant. The *rev3Δ* mutant on its own is not sensitive to HU, and the double mutant with *pol32Δ* does not show enhanced sensitivity. *pol31-W417* efficiently suppresses the growth defect on 25 mM HU in the absence of Rev3 (see triple mutant).

the *pol31-T415* mutation, and the suppressor *pol31-W417* partially restored Pol2 recovery in *pol32Δ* cells (Fig 5D). Thus, the HU sensitivity of cells lacking Pol32 may not simply reflect reduced Pol3 levels, but an indirect effect on replisome stability during HU arrest, triggered by reduced levels of Pol δ.

### BIR is not restored in *pol32Δ* strains by enhanced Pol31–Pol3 interaction

Given that *pol3-W417* could confer at least moderate rescue of fork recovery after HU induced fork arrest, and complements *pol32Δ* sensitivity to other DNA damaging agents, we next asked whether it

suppresses the *pol32Δ* defect in BIR. BIR is a recombination-dependent pathway for fork recovery that is largely dependent on Pol32 (Lydeard et al, 2007). To see if the *pol31-W417* suppression of *pol32Δ* restores BIR, we used a well-characterized BIR assay in both the *pol31-T415* and *pol31-W417* mutants, with *pol32Δ* as (negative) control (Fig 7A). BIR in this case is triggered by an induced HO-endonuclease DSB on one of two unrelated chromosomes, which share a stretch of homology that promotes strand exchange and elongation. Successful BIR is strongly compromised in the absence of Pol32 (Lydeard et al, 2007) (Fig 7B and C). However, in this assay, the *pol31-T415* and *pol31-W417* mutants were both less

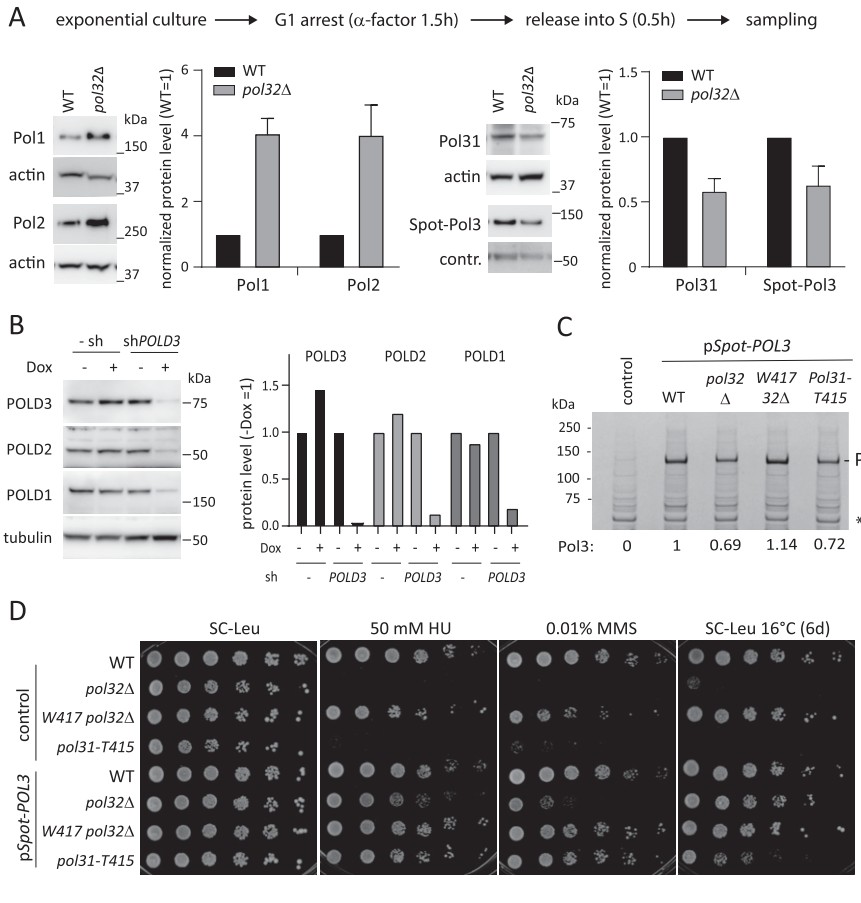

**Figure 5. Loss of Pol32 or POLD3 reduces levels of other Pol δ subunits and a second copy of POL3 suppresses *pol32Δ* and *pol31-T415A* sensitivities.**
**(A)** Loss of Pol32 down-regulates Pol δ levels but up-regulates Pol1 and Pol2 in budding yeast. Pol2-13Myc–tagged wild-type (GA-5050) and *pol32Δ* (GA-6292) cells were synchronized in G1 by α-factor then released to S phase for 30 min, before total protein extraction. Total protein extracts were subjected to Western blots probed for Pol1 (Pol α), Pol2-13Myc (Pol ε), Pol31 (Pol δ), and the Spot-tag Pol3 (Spot-Pol3). The pSpot-tagged Pol3 is expressed from a single copy plasmid pSpot5 or p*Spot5-POL3* (GA-4619 and GA-4761) which shows no difference from the wild-type strain for growth at any temperature, nor for HU or MMS (panel D). Intensity of each band was quantified and normalized to actin or a control band (for Pol3). Pol31 and Pol3 reproducibly show lower levels in *pol32Δ*. The graphs show means and standard deviation from the mean from four (Pol1, Pol2, and Pol31) or three (Pol3) biological replicates in each case. For standardization, WT signal is normalized to 1. **(B)** POLD3 knockdown by shRNA results in loss of other Pol δ subunits in mammalian cancer cells. Dox-inducible shPOLD3 was transfected into Jurkat cells. Non-transfected parental and shPOLD3 transfected cells were treated with or without 1 μg/ml Dox for 3 d. Total protein sample from each culture were subjected to Western blot probed for POLD3, POLD2, POLD1, and tubulin. Intensity of Pol δ subunits was normalized with tubulin signal, then plotted as no Dox signal = 1. Data were reproducible in three independent replicates. **(C)** Pull-down of Spot-Pol3 shows that it is destabilized in the *pol32Δ* or *pol31-T415* background, whereas the double *pol32Δ pol31-W417* mutant restores Spot-Pol3 levels. Cells with the indicated genetic background express N-terminally tagged Spot-Pol3 or control (Spot peptide alone) from a single copy plasmid pSpot5 or p*Spot5-POL3* (GA-4619 and GA-4761). Synchronized cell populations were lysed and the Spot-tagged Pol3 was recovered by an excess of Spot-Trap Magnetic Agarose beads. Recovered protein was denatured, and equal amounts of each sample were loaded on a 4–20% polyacrylamide SDS gel. Instant Blue dye staining was quantified by scanning, and Pol3 was normalized to the nonspecific control band. **(D)** Serial dilution (5×) drop assays on the indicated selective media was performed for strains with the indicated genotype bearing a single copy plasmid expressing either the Spot tag alone (control) or the N-terminally tagged Spot-Pol3 protein. Plates contained the indicated concentration of MMS or HU, or neither, and were incubated at 16°C for 6 d. All other plates were incubated for 3 d at 30°C before imaging. The single copy p*Spot-POL3* vector suppresses the MMS, HU, and cold-sensitivity of *pol32Δ* and *pol31-T415*.

efficient at BIR than wild-type cells, and the *pol31-T415* allele, which phenocopies *pol32Δ* in other assays, was far less compromised for BIR than cells lacking Pol32 (Fig 7B and C). Moreover, *pol31-W417* was unable to compensate for *pol32Δ* in the *pol32Δ pol31-W417* double mutant. Similarly, the pSpot-Pol3 restored only minor function in the *pol32Δ* background (Fig 7D and E), in a similar BIR assay, triggered by an HO-induced break.

In conclusion, the stabilized Pol3–Pol31 interaction was able to compensate for the loss of Pol32 on MMS and on HU, and—very similarly—a second copy of Pol3 was able to compensate for the impaired growth of *pol32Δ* strains on MMS, HU and at 16°C. On high concentrations of HU; a stable Pol3–Pol31 interaction was needed to stabilize the replisome at forks stalled. However, both *pol31-T415* and *pol31-W417* compromised BIR on their own, and neither *pol31-W417* nor elevated Pol3 levels complemented *pol32Δ* for BIR (Fig 7). With this analysis, our work defines distinct roles of the Pol δ holoenzyme and of its two smaller subunits, in different types of repair. Both Pol31 and Pol32 play important roles in the resumption of replication by BIR, yet these roles are distinct from the redundant role the subunits play in ensuring holo-Pol δ abundance on HU and MMS.

# Discussion

Many of the pathways that control replication fork recovery under control of Mec1/ATR make use of the lagging strand polymerase, DNA Pol δ (Maga & Hübscher, 2008; Hübscher & Maga, 2011). In budding yeast Pol δ is a heterotrimeric enzyme consisting of the catalytic subunit Pol3 (human POLD1/p125), Pol31 (POLD2/p50), and Pol32 (human POLD3/p66). The human and fission yeast complexes also contain a smaller regulatory subunit POLD4/p12. The DNA polymerase processivity factor, PCNA, interacts through the multiple PIP sites in Pol δ, with a strong essential interaction through the CTD of Pol3 and possibly, as well, the C terminus of Pol32 (Johansson et al, 2004; Acharya et al, 2011).

The second and third Pol δ subunits, Pol31 and Pol32, are needed for DNA damage or replication stress survival, particularly in PRR pathways for the repair of alkylated bases. Consistently, Pol31 and Pol32 were found as core subunits of the DNA Pol ζ (Rev3), which mediates error-prone TLS (Johnson et al, 2012; Makarova et al, 2012). However, we show here that Pol3 and not Rev3 is important for TLS and survival of alkylation damage by MMS in the absence of Pol32

A

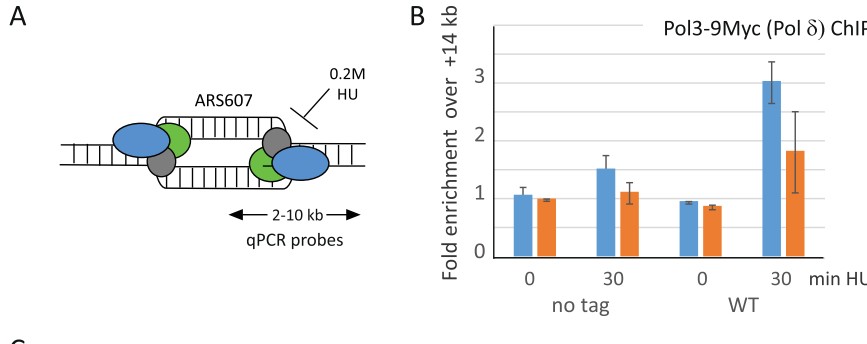

B

Pol3-9Myc (Pol δ) ChIP

C

Pol1-3HA (Pol α) ChIP

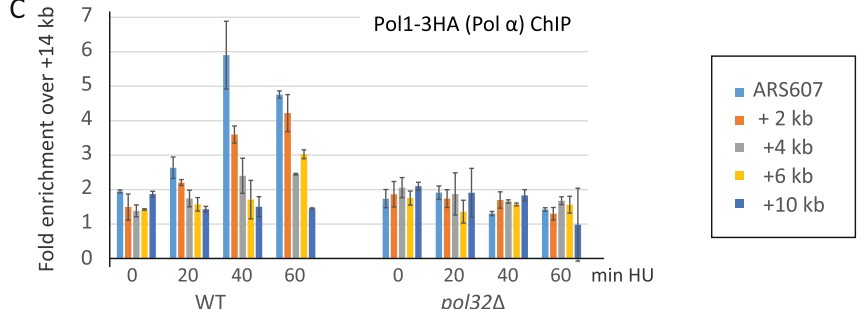

D

Pol2-13Myc (Pol ε) ChIP

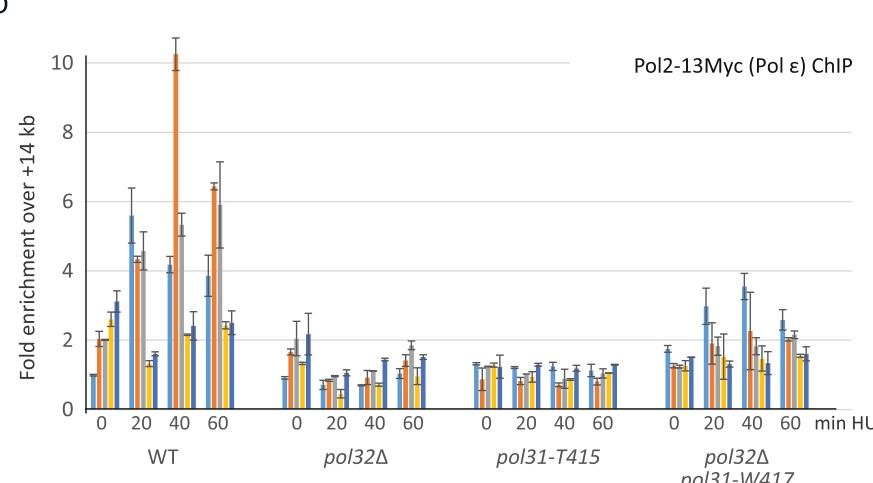

**Figure 6. DNA polymerase binding at HU-stalled replication forks is sensitive to *pol32Δ*.**
**(A)** Schematic representation of HU-stalled fork around ARS607, an efficient early firing origin on Chr VI. Synchronization and release protocols are in the Materials and Methods section. **(B)** DNA pol δ catalytic subunit Pol3 is detected at stalled forks in wild-type cells. Myc-tagged Pol3 is enriched over the non-tagged control at ARS607, and at +2 kb, as demonstrated by CHIP. **(C)** Probes for qPCR are color-coded (C). CHIP was performed in three independent experiments and qPCR for the indicated probes was performed in triplicate for each biological replicate. Values are normalized to the signal at ARS607 + 14 kb, which is halfway to the next origin of replication. **(C)** DNA pol α catalytic subunit Pol1, is destabilized by *pol32Δ* at a stalled replication fork. **(B)** As (B), a C-terminally 3-HA-tagged Pol1 is found enriched at stalled replication fork near early firing ARS607 on 0.2 M HU, and it is lost in an isogenic *pol32Δ* strain. **(B, D)** As (B), but for Myc-tagged Pol2, the catalytic subunit of Pol ε. Identical CHIP assays were performed in strains bearing *pol32Δ*, *pol31-T415* and double *pol32Δ pol31-W417* mutations. The T415 mutation, like *pol32Δ*, disrupts Pol ε enrichment, whereas *pol31-W417* partially compensates for its recovery at stalled forks.

(Fig 3). Consistently, a study in chicken DT40 cells showed that Pol δ contributes to TLS independently of Pol ζ (Siebler et al, 2014; Hirota et al, 2015). Notably, the inactivation of the proofreading activity of POLD1 restored TLS activity in POLD3-deficient chicken DT40 cells, suggesting that Pol δ also mediates TLS (Hirota et al, 2016).

Consistent with key functions of Pol32 in dealing with DNA fork-associated damage, we show that cells lacking Pol32 show cold-, HU-, and MMS sensitivity, and that this deficiency can be suppressed by a point mutant in Pol31 (*pol31-W417*) that enhances interaction between Pol31 and the Pol3 C-terminal CysB element. This mutation restores the level of DNA Pol3 and Pol31, which are compromised by the loss of Pol32. Similarly, the neighboring *pol31-W417* mutation phenocopies the *pol32* null allele, in MMS- and HU survival assays, but not for BIR. The impact of the gain-of-function mutation on replisome stability at forks stalled by high dose HU lies in between its effects on MMS and in the BIR assay. Finally we note

that suppression of *pol32Δ* on MMS requires the error-free lesion bypass pathway that depends on Rad5, Mms2, and Ubc13, whereas suppression of *pol32Δ* on HU does not. In any case, we can attribute the hypersensitivities linked to Pol32 ablation, to the instability of Pol3 and Pol31, and reduced steady state levels of both proteins (Fig 5A).

Not all functions of Pol32 are complemented by enhancing the Pol31-Pol3 interaction, and most notably, it seems that both Pol32 and Pol31 contribute to efficient BIR, and this appears not simply reflect the stabilization of Pol3 levels. Previous studies have shown that DNA Pol δ is critical for elongation after strand invasion during homologous recombination (HR [Maloisel et al, 2008]), which is also the case during BIR (Brocas et al, 2010). A four–amino acid C-terminal truncation of Pol3 results in shorter track length after strand invasion, but does not fully compromise HR, and neither Polη nor Rev3 are involved in this type of repair. Interestingly, the same Pol3

## A

**Break-induced Replication assay (1)**

## C

| strain | rate of HO-cut repair by BIR | Survival rate (%) (WT=100) |
|---|---|---|
| WT | 0.0644 +/- 0.023 | 100 |
| pol31-T415A | 0.0342 +/- 0.0144 | 64.58 +/- 13.94 |
| pol31-W417A | 0.0237 +/- 0.0098 | 54.34 +/- 25.91 |
| pol32Δ | 0.00044 +/- 0.0003 | 2.457 +/- 0.42 |
| pol32Δ pol31-W417A | 0.0012 +/- 0.0008 | 5.678 +/- 1.90 |

## D

**Break-induced Replication assay (2)**

## B

## E

**Figure 7. Break-induced replication (BIR) assays are sensitive to *pol31-W417* and *pol31-T415* and neither can suppress the BIR deficiency of *pol32Δ*.**
**(A)** The *CAN1* gene is replaced by *URA3* and the *HPH* by clonNAT in GA-8997 (gift of J Haber). After HO endonuclease break induction, a homology-driven repair event between the two chromosomes is mediated by BIR, restoring uracil prototrophy and clonNAT-sensitive viable colonies. In the GA-8997 strain, the donor 5'-UR- is 33 kb away from Tel-VL and the recipient 3'-RA3 is 39 kb away from Tel-IXL. **(B)** BIR efficiency was scored in GA-8997 which is wild type for all subunits of Pol δ, and in the same background bearing the indicated *pol31* and *pol32Δ* mutations. Each assay was quantified in triplicate and each assay was replicated at least three times. BIR is strongly affected by *pol32Δ* and is slightly reduced by both *pol31*-T415A and *pol31*-W417A mutations. Importantly, only minor suppression of *pol32Δ* by *pol31-W417A* can be detected by an HO-induced BIR assay. We confirmed that HO cut efficiency was comparable in all strains, and repair efficiency of induced HO cleavage was normalized cut efficiency. **(C)** BIR efficiency and its normalized value in WT as 100 was shown in the table and plotted (right). **(D)** Schematic representation of BIR assay in Fig 7E. **(E)** The ectopic expression of *POL3* (pSpot-POL3) on a CEN plasmid does not suppress the loss of BIR in *pol32Δ*. WT (GA-5998) and *pol32Δ* (GA-5999) cells were transformed with a control (pSpot) or Spot-POL3 expressing plasmid (pSpot-POL3). **(A, B)** Cells were cultured in SC-LGG-Leu, and tested for BIR assay as in panels (A, B) above, except *CAN1* was scored for the successful BIR. Mean of BIR efficiency from three (WT) and four (*pol32Δ*) biological repeats was plotted. Error bar represents standard deviation of the mean.

---

C-terminal truncation is synthetic lethal with *pol32Δ* and can be suppressed by a mutation at amino acid 358 (K to I or K to E [Brocas et al, 2010]). Pol31-K358 sits on the same solvent exposed loop as T415 and W417, and like these two residues, contributes to Pol31–Pol3 interaction (Fig 2 [Brocas et al, 2010]). Here we distinguish the different Pol32-dependent pathways of repair that can be suppressed by improving Pol31–Pol3 interaction. We show that there are different roles of Pol32 in BIR, PRR, and fork recovery from high concentrations of HU. Importantly, our results underscore a unique pathway of Pol32 action in BIR, which cannot be suppressed by improved Pol31–Pol3 interaction.

In mammalian and *Drosophila* cells, the loss of either POLD3 or POLD2 leads to the degradation of the other subunits of holo-Pol δ (reviewed in Fuchs et al [2021], Fig 5B). We show a similar drop in budding yeast Pol3 levels upon loss of Pol32, and/or introduction of the mutation *pol31-T415* (Fig 5A and C). The instability of Pol3 was restored by combining *pol32Δ* with *pol31-W417*, which stabilizes Pol3 levels (Fig 5C), whereas the expression of one extra copy of wild-type *POL3* (pSpot-Pol3) also could suppress the MMS-, HU- and cold-sensitivity of *pol32Δ* (Fig 5D). Thus, steady-state levels of holo-Pol δ are crucial in face of replication stress, whether triggered by alkylation or HU. The Pol δ threshold may also enable replisome

maintenance at forks acutely stalled on HU, yet there is an additional Pol32 role in BIR (and presumably HR) beyond the maintenance of holo-Pol δ levels, given that *pol31-W417* could not restore, and instead compromised, BIR.

These observations are relevant to the overexpression and up-regulation of POLD2 and POLD3 in human cancers. Given the conservation between structure and sequence of Pol δ subunits, we propose that the up-regulation of POLD2 or POLD3 stabilizes POLD1, and potentially improves its ability to cope with replication stress. Under the persistent replication stress that is found in precancerous lesions (Gorgoulis et al, 2005), POLD2 may become essential for cell survival. We predict that the inhibition of Pol δ in cancer cells will sensitize tumors to chemotherapy by reducing damage survival. Our data show that the increased affinity of pol31-W417 for Pol3 prevents destabilization and degradation of Pol3 that occurs in the absence of Pol32. Whereas pol31-W417 can also confer a more stable interaction of Pol31 with the TLS polymerase, Rev3, yeast Rev3 is not implicated in survival on HU (Fig 4).

Based on the crystal structure of the human orthologue of Pol31-Pol32N-term (Baranovskiy et al, 2008), which was confirmed by more recent crystal structures (Jain et al, 2019; Lancey et al, 2020), we mapped the novel *pol31-T415* and *pol31-W417* mutations to a highly conserved, solvent exposed loop on the surface of Pol31, which

forms the interface between Pol3 and Pol31 (Fig 1C). The *pol31-T415* mutation results in a loss of interaction between Pol31 and Pol3-CTD, whereas *pol31-W417* enhances it (Fig 2). A stable Pol3–Pol31 interaction impacts the stability and function of Pol δ under a variety of conditions. First, we note that the Pol31 T415/W417 loops binds the Fe-S cluster CysB of Pol3. Given that HU not only reduces dNTP levels, but destabilizes Fe-S clusters in general (Huang et al, 2016), it may weaken the Pol31–Pol3 interaction in a dose-dependent manner. Thus, stabilizing this interface (through pol31-W417) may help the Fe-S cluster resist disruption on HU. Indeed, we found that Rad18, Rad5, and Mms2 are necessary for growth of the *pol32Δ* strain on low HU, and that *pol31-W417* partially suppresses this HU sensitivity in the absence of Rad18, Rad5, or Mms2. However, *pol31-W417* could not suppress the growth defect at higher HU concentrations (e.g., 100 mM HU) without Rad18, Rad5, or Mms2 (data not shown). This is not the first instance in which mechanistic differences between low and high (20 mM versus 100 mM) concentrations of HU were observed (Hustedt et al, 2015). This observation suggests either that Rad6-Rad18 and Mms2 become necessary at higher HU concentrations, or that the higher concentrations of HU leads to more extensive Fe-S protein disruption (Huang et al, 2016), such that the *pol31-W417* mutant can no longer stabilize Pol3 in the absence of Pol32.

Importantly, the *pol31-W417* mutant does not compensate for the impact of *pol32Δ* during BIR, which is assayed in the absence of HU. Moreover, the Pol3 T415A and W417A mutations both partially compromise BIR. Previous work suggested that the role of Pol32 in BIR is mediated primarily through the binding of its N terminus to Pol31 (Lydeard et al, 2007); however, this interaction is not compromised by mutations in the Pol3-binding loop of Pol31 (Fig 3C). Thus, our work identifies unique roles for Pol32, and Pol31, in efficient BIR. Probing this and the role of ubiquitin ligases such as Rad6/Rad18 in Pol δ activity will be a focus of future studies.

# Materials and Methods

### Yeast culture, strains, and plasmids

The yeast strains and plasmids are described in Tables S1 and S2, respectively. Mutant *pol31* yeast strains were prepared by plasmid shuffling. If not stated otherwise cells were cultured at 30°C in YPAD (1% Bacto yeast extract, 2% Bacto peptone, 0.004% adenine, 2% glucose) medium using standard procedures, unless otherwise indicated.

A PCR-based epitope tagging method was used to tag *POL1*, *POL2*, *POL31*, and *POL3* genes by chromosomal integration of PCR-amplified cassette in the KSC106 or W303 yeast background used for genetic analyses. The correct integration of the cassette was verified by PCR as well as checking the expression of the Myc- or HA tag by Western blotting. To see if the hypersensitivity of the isolated ts/cs *pol31* mutants reflects polymerase destabilization, we tagged *POL3* with the N-terminal Spot epitope in the single copy pSpot5 CEN plasmid (ChromoTek). These cells were viable and the tag did not compromise Pol δ function, whereas C-terminally 9-Myc or FLAG-tagged Pol3 was lethal when combined with *pol32Δ* (Fig S4B).

### Yeast two-hybrid analysis

For yeast two-hybrid analysis, wild-type and indicated mutants of Pol31, and the full-length Rev3 were fused to the lexA DNA binding domain of pGAL-LexA (Bait). The WT and indicated mutants of Pol31, Pol32, PCNA and the C-terminal CysA and CysB domains of Pol3 were also fused to the B42 transcription activation domain of pJG4-7 (Prey). Liquid β–galactosidase assays were performed as previously described (Hegnauer et al, 2012) using isogenic wild-type yeast strain (GA-1981) or *pol32Δ* mutant (GA-6292) containing the lacZ reporter pSH18-34, the bait, and the prey. Exponentially growing, glucose-depleted cells were exposed to 2% galactose for 6 h to induce the fusion proteins, and protein–protein interactions were detected by the quantitative β-galactosidase assay of cell extracts. Three to four independent transformants were analysed for each interaction. Expression of all the fusion proteins was confirmed by Western blot analysis.

### Pull-down assays

The scheme in Fig S2B was used for pull-down assays. Yeast cell extracts containing over-expressed HA-Pol31 derivatives (B42/WT-*POL31*/*pol31-T415*/*pol31-W417*, plasmids #1493, #3062, #3063, #3064) or equivalent LexA-Pol31 derivatives (#2686, #2856, #2855, #3816, Fig S2C) were supplemented with protease inhibitors (Complete protease inhibitors; Roche), and added to *Escherichia coli*–expressed MBP-Pol3-CTD bound to Amylose (Sigma-Aldrich) beads: CysA + CysB (Pol3 991-1097) or only CysB (Pol3 1029-1097). The yeast extracts were derived from cultures grown either at 30°C or 16°C, as indicated, and pull-down was carried out at 4°C. The amount of Pol3 CysA + CysB protein used was determined by UV-imaged Criterion gels (Bio-Rad) and the binding of Pol31 to Pol3-CTD was analysed by Western blot using anti-HA antibody. Equal expression of the Pol31 constructs was checked by Western blotting for the epitope-tagged protein in equivalent amounts of whole cell extracts (Fig S2E and F). The coupling of the MBP fusions to beads was performed in the lysis buffer (50 mM Hepes, pH 7.5, 20 mM NaCl, 1 mM EDTA, and 0.1% Triton X-100, with the Roche protease inhibitor cocktail) for 1.5 h at room temperature, and washed three times with lysis buffer before use.

Whole cell extracts were derived from exponentially growing yeast cells of the indicated genotype expressing the indicated Pol31 fusions (~1 × $10^7$ cells/ml, 200 ml). Cells were pelleted and washed once with ice-cold PBS, resuspended in 0.8 ml lysis buffer, and subjected to beat beating with Zirconia beads, 6.5 Hz, 60 s, four times at 4°C. Cell lysate was clarified by centrifugation 12,000*g*, 5 min at 4°C. 150 μl of total cell lysate was incubated with MBP-Pol31 bound to Amylose beads for 1.5 h at 4°C with constant rotation. The beads were washed three times with excess lysis buffer containing 0.1% Triton X-100 and 20 mM NaCl at 4°C. Bound proteins were eluted from the beads, the protein sample was denatured and boiled in 1× NuPAGE sample buffer, and analysed by NuPAGE. Further analysis was by UV imaging of the Criterion gels and Western blotting.

For the quantitation of Pol3 levels (Fig 5), strains expressing N-terminally tagged Spot-POL3 or control (Spot peptide alone) from a plasmid, were grown to OD600 = 0.8 (exponential growth) in SC-leucine. Equal numbers of cells from each culture were lysed and the total cell lysate was incubated with an excess of Spot-Trap Magnetic Agarose beads for 2 h at 4°C. The Spot-tagged Pol3 was recovered by from the agarose beads and denatured. An equal amount of each sample was loaded on 4–20% polyacrylamide SDS gel, which was probed with Spot VHH nanobody (ChromoTek)

followed by anti-Llama-IgG-HRP conjugate (BRTHYL) in Fig 5A, or stained with Instant Blue dye (Abcam) in Fig 5C. Pol3 levels were normalized to an invariant non-specific control band.

## Cell recovery, drop assays, and checkpoint activation

All methods for checkpoint activation by phospho-shift analysis of Rad53 phosphorylation, for cell sensitivity to DNA damaging agents by drop assay, and cell recovery assays from MMS or HU after α-factor synchronization were performed essentially as described in Hustedt et al (2015). For liquid survival assays, overnight cultures were diluted to OD600 = 0.15 and grown for 3 h, then synchronized with α-factor in G1 and released into 0.2 M HU containing YPAD. After the indicated time points relevant dilutions were plated onto fresh YPAD plates and colonies were counted after 3–4 d. Survival is defined as the fraction of indicated doses compared to the untreated control (0 h) normalized to the survival of WT cells for each time point.

For serial dilution drop assays, overnight cultures were diluted to a starting density of OD600 = 0.5 and 2 $\mu$l drops of 10-fold dilutions were plated on YPAD plates or on an appropriate selective medium containing freshly diluted concentrations of MMS or HU as indicated. Where indicated, 5× dilution series were used. Incubation of plates was usually at 30°C, but where indicated it was 16°C or 20°C.

To monitor Rad53 activation by Western blot, exponentially growing cells were synchronized in G1 with α-factor and released into 0.2 M HU, or 0.03% MMS containing YPAD media at 30°C for 0.5–1 h. To monitor cs, pol31 mutants were released into YPAD media at 16°C and grown for 1 h at 16°C. For Western Blot analysis, lysates were prepared using silica beads and urea buffer and subjected to 7.5% SDS–PAGE, transferred onto poly(vinylidene fluoride) membrane and detected with anti-Rad53 antibody (sc-6749; Santa Cruz).

## ChIP analysis

ChIP experiments were performed as described (Cobb et al, 2003, 2005) using GA-5050, GA-6292, GA-10007, GA-10008, GA-4796, GA-6123, and GA-6290. Monoclonal anti-HA (F-7; Santa Cruz) was used to precipitate HA-tagged DNA pol α, and anti-Myc (9E10) to precipitate Myc-tagged Pol2 and Pol3. BSA-saturated anti-mouse IgG-Dynabeads (Thermo Fisher Scientific) were coupled to the monoclonal antibodies and incubated with the cell extract for 2 h at 4°C. Antibody-coupled Dynabeads without each strain are averaged over three independent experiments with real-time PCR performed in duplicate (error bars indicate standard error of the mean). Absolute fold enrichment at ARS607 was calculated as follows: for each time point the signal from the antibody-coupled Dynabeads was divided by the signal from the BSA-coated Dynabeads, after both signals were first normalized to the signal from the input DNA. Finally, the relative enrichment for ARS607 was obtained by normalizing the absolute enrichment at the indicated probe near ARS607 for each time point to the absolute enrichment at a locus 14 kb away from ARS607.

## BIR assay

Based essentially on the published protocols in Lydeard et al (2010) and Anand et al (2014) we used the BIR-tester strain, GA-8997 and derivatives of it (Table S1), in which the CAN1 gene is replaced by URA3 and the HPH by clonNAT (gift of J Haber). A galactose-inducible HO endonuclease generates a DSB within the UR3-NAT locus on Chr V. Repair of the DSB by BIR results in URA+ and clonNAT-sensitive viable cells. In GA-8997 the donor 5′-UR3 is 33 kb away from Telo-VR and the Recipient 3′-RA3 is 39 kb away from Telo-IX-R. All strains from −80°C were directly streaked out on YPAD + clonNAT plates to recover only cells with intact HO-sites. Then cultures were grown in YPA-2% Raffinose for 6 h to 10$^7$ cells/ml and 10-fold serial dilutions were plated each on YPAD (to obtain total cell count), on SC-ura (to score background level URA+) and on YPA-galactose to induce the HO endonuclease. Cell viability after HO-endonuclease-induction was derived by dividing the number of CFUs on YPA-Galactose by that on YPAD. Rate of BIR repair was determined by the product of URA+ CFU over YPAD CFU times the number of URA+ CFU over YPA-Gal CFU. The BIR assay in Fig 7E was conducted in the same way, except that CAN1 positive colonies (L-canavanine sensitivity) were scored for successful BIR in strains GA-5998 and GA-5999.

## Homology modeling of the Pol31–Pol32N complex

HHPRED (Söding et al, 2005) was used to search the Protein Data Bank (www.rcsb.org) for Pol31 and Pol32 orthologues of known structure. For both Pol31 and Pol32, the crystal structure of the human P50-P66(N) complex (PDB 3 E0J [Baranovskiy et al, 2008]) was the top hit which gave high scores for Pol31 matching P50 (probability = 100%, E-value = 1.2 × 10$^{-66}$, identities = 32%, Pol31 residues 19–486) and Pol32 matching P66(N) (probability = 99.93, E-value = 3.4 × 10$^{-26}$, identity = 13%, Pol32N residues 1–127). HHPRED pairwise sequence alignments for Pol31/P50 and Pol32/P66(N) were locally corrected, then merged and used for modeling the heterodimeric complex using the modeller software (Sali & Blundell, 1993) together with the 3E0J crystal structure as 3D template (chains A, B). Pol31 loops longer than 10 residues without structural information in the 3D template were excluded from modeling calculations. 250 models for the Pol31–Pol32N complex were generated and the best model was chosen according to the lowest values for the modeller objective function and the quality of the Ramachandran plot.

# Supplementary Information

# Acknowledgements

We thank the J Haber laboratory for useful strains, P Plevani for Pol1 antibody, P Burgers for Pol31 antibody, and Cleo Tarashev for the assistance on BIR assay. We are grateful to all members of the Gasser laboratory for helpful suggestions for this study. We also acknowledge the generous financial support of the Swiss National Science Foundation grant number 31003A-176286 to SM Gasser and the Swiss Cancer League grant 4167-02-2017 to SM Gasser and N Delgoshaie. We thank the Novartis Research Foundation for continued core support of the Friedrich Miescher Institute for Biomedical Research (FMI).

## Author Contributions

K Shimada: conceptualization, resources, data curation, formal analysis, validation, investigation, visualization, methodology, and writing—original draft, review, and editing.

M Tsai-Pflugfelder: conceptualization, resources, data curation, formal analysis, validation, investigation, visualization, methodology, and writing—original draft.

ND Vijeh Motlagh: conceptualization, resources, data curation, formal analysis, validation, investigation, visualization, methodology, and writing—original draft.

N Delgoshaie: conceptualization, resources, data curation, software, formal analysis, funding acquisition, validation, investigation, visualization, methodology, and writing—review and editing.

J Fuchs: data curation, formal analysis, validation, investigation, visualization, methodology, and writing—review and editing.

H Gut: data curation, software, formal analysis, investigation, visualization, and methodology.

SM Gasser: conceptualization, resources, data curation, formal analysis, supervision, funding acquisition, visualization, methodology, project administration, and writing—original draft, review, and editing.

## Conflict of Interest Statement

The authors declare that they have no conflict of interest.

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
