## [Reviewer comments · Life Science Alliance]

Life Science Alliance

The stabilized Pol31-Pol3 interface counteracts Pol32 ablation with differential effects on repair

Susan Gasser, Kenji Shimada, Monika Tsai-Pflugfelder, Niloofar Vijeh Motlagh, Neda Delgoshai, Jeannette Fuchs, and Heinz Gut

DOI: <https://doi.org/10.26508/lsa.202101138>

Corresponding author(s): Susan Gasser, Friedrich Miescher Institut for Biomedical Research

Review Timeline:

Submission Date:	2021-06-17
Editorial Decision:	2021-06-21
Revision Received:	2021-06-23
Accepted:	2021-06-24

Transaction Report:

Please note that the manuscript was previously reviewed at another journal and the reports were taken into account in the decision-making process at Life Science Alliance.

Referee #1 Review

Report for Author:

This manuscript reports a mutagenesis screen of POL31 in *S. cerevisiae*. Several phenotypic groups were identified and the focus is on two mutations that define: (1) weakening of the interaction between Pol3p and Pol31p. This mutant, as would be expected, is toxic in the context of pol32-deletion, which is known to contribute to the stabilization of the polymerase delta complex. (2) a mutant that appears to increase the strength of the interaction with Pol3p. This mutant reverses the genotoxic sensitivity (and cold sensitivity) of the pol32-deletion mutant. This is consistent with the role of Pol32 in stabilizing the polymerase delta complex. The fact that this same mutant does not rescue the defect of pol32-deletion in a BIR assay strongly suggests additional roles for Pol32p (as suggested by other studies), but these are not elucidated. In addition there are a couple of interesting observations such as the fact that destabilizing polymerase delta complex impacts on the retention of Pol1 and Pol3 at HU arrested replication forks. Again these are not mechanistically followed up.

Comments.

The terminology EF463 is unclear. When first introduced (page 5 results) it should be explained.

On page 6. The two mutants do not formally define the conserved domain, they do allow its function to be probed.

Page 9. Why is it interesting that the sensitivity is more pronounced when cells are exposed to both MMS and HU?

Page 9. Does the increased levels of Pol1 and Pol2 in the pol32-deletion simply reflect an increased time in S phase, where the genes are transcribed?

I do not see any reference to figure 5B, which reports that human POLD1 and POLD2 are, as has been reported previously, destabilized upon POLD3 knockdown.

Referee #2 Review

Report for Author:

In the study, Shimada, et al, identified and characterized novel mutations in POL31, encoding one of the two regulatory subunits of DNA polymerase Pol delta holoenzyme in budding yeast. Specifically, they described pol31-W417 mutation that suppresses cellular sensitivity to MMS, HU and low temperature in pol32 deletion mutant. This is in contrast to the closely related pol31-T415 that seems to phenocopy pol32 deletion. They performed yeast 2-hybrid and in vitro pull-down experiments that suggest pol31-T415 mutation compromises interaction between Pol31 and Pol3 while pol31-W417 improves such interaction in the absence of Pol32. They further presented data that explores the molecular mechanism underlying the phenotypic observations. Importantly they found that the presence of Pol32 as well as Pol31 - Pol3 interaction enhance the stability of Pol delta. This study provides further insight into the function of the Pol31 subunit.

Comments:

1. In the yeast two-hybrid and in vitro pull-down experiments, the authors used two versions of the extreme c-terminal fragments of the Pol3 protein (1097) containing only the CysA + CysB (~100 aa) or CysB-only (~60 aa) domains. Importantly, the two Pol3 behaved differently. The interaction of CysA+B and Pol31 was not affected by POL32 deletion while CysB - Pol31 interaction was. W417 mutation had no significant impact on CysA+B - Pol31 interaction, while showing dramatically positive influences on CysB - Pol31 interactions. The in vitro pull-down data presented in Figure 3E is less interpretable, as it lacks the WT control at 30 degree. These differences between different constructs raise the concern whether the truncated Pol3 represents the behavior of the full-length protein. Furthermore, the authors have not tested the protein stability of the mutant Pol31, which further complicates the interpretation of the data.

2. Figure 2E, it is not clearly what the authors mean by 16 vs 30 degrees for the pull-down experiments. I suppose it is the temperature at which the cells were cultured before lysed for in vitro pull-down. If this is the case, they have not sufficiently laid out the rationale for such experimental design and discussed the difference observed, since the cell lysate is eventually incubated with a recombinant Pol3-CTD fragment in vitro at 4 degree.

3. Figure 3, rev3 \square showed no sensitivity at the treatment conditions. Is it due to the redundant role of Pol eta in TLS? If this is the case, the data is not sufficient in testing whether the TLS branch

contribute to W417 suppression.

4. Figure 3C. The interaction between rad18 and pol32 is a little odd considering both rad5 and mm2 show strong synergy with pol32. The authors may need to double check the strains and maybe repeat the experiment.

5. Figure 5C. Since the changes of the expression level of Pol3 is relatively minor, this experiment need repetition and quantitation with error bar.

Referee #3 Review

Report for Author:

This is a mutational study of Pol31, the second subunit of DNA polymerase delta (and DNA polymerase zeta). A previous study of Pol31 identified a number of cs/ts mutations and both genetic and physical interactions with several replication/repair pathway proteins, including Mgs1, Rad52, and Srs2. This follow-up study focuses in on two Pol31 mutants. One mutation, T415A, severely compromises interactions with the catalytic subunit Pol3, whereas the other mutation, W417A, shows modestly increased interactions with Pol3, but more importantly for the paper, it partially suppresses the deleterious phenotypes caused by deletion of Pol32, the third subunit of Pol-delta (and zeta). Control studies with rev3-deletion mutants show that the phenotypes are primarily pol-delta related. It is a potentially interesting study, although it may be more suited for a specialized readership than for the DNA repair/replication community in general. Unfortunately, the study is quite uneven in quality, and renewed studies are required to elevate the whole to that of consistent high quality.

Major comments:

1. The data show that the W417 mutation partially suppresses the various phenotypes of the pol32 deletion and moreover that, with the exception of BIR, this suppression is largely pathway independent. The authors suggest that this is the effect of an increased interaction between Pol3 and pol31-W417A. However, it may well be possible that this mutation actually alters the biochemical properties of the two-subunit complex, such that it shows more robust activity. Wild-type Pol3-Pol31 is compromised for several activities shown by the three subunit enzyme. The Pol31 mutation may enhance some, but not all of them. One interesting finding in this study is that pol31-W417 partially suppresses the PRR, but not the BIR defects of pol32-delta. Does it relate to Pol3-Pol31 levels or its activity?

This study would be more comprehensive if it were complemented with biochemical studies of the mutant Pol3-31 complex.

2. Figure 5, Figure S4: It is surprising that authors used C-terminal tagging of Pol3 in several experiments, even while they are cognizant of the fact that this tagging is deleterious for Pol3 function. In contrast, N-terminal tagging (used by others) does not appear to show defects. Without proper tagging, these studies hold little value. But see point 4.

3. There is a curious misquoting of the primary literature in this paper. For instance, 4 papers from 2006-2015 are quoted for the monoubiquitination of PCNA by Rad6/18, while the landmark 2001 Nature paper by Hoege et al. is completely missed. Similarly, the identification of CysA and CysB

motifs and the placement of the Fe-S cluster has several references, however this analysis originated from a Netz et al. 2011 Nat Chem Biol paper which was not referenced.

4. The suppression of pol32-delta by an extra copy of sPot-Pol3 is interesting. However, this altered, tagged form of Pol3 may not be under the same type of regulation as wild-type Pol3. To obtain a clean result, the experiment should be repeated with an extra, integrated, copy of Pol3, containing its proper regulatory sequences.

Minor comments:

5. Introduction: An inordinate amount of space is taken up in describing the results of the study. The three last paragraphs could be condensed into one small paragraph.

6. The compensatory upregulation of Pol1 and Pol2 in pol32-delta is interesting. Can some of this be explained by the increased S-phase residence of the mutant cells?

Referee #1:

This manuscript reports a mutagenesis screen of POL31 in *S. cerevisiae*. Several phenotypic groups were identified and the focus is on two mutations that define: (1) weakening of the interaction between Pol3p and Pol31p. This mutant, as would be expected, is toxic in the context of pol32-deletion, which is known to contribute to the stabilization of the polymerase delta complex. (2) a mutant that appears to increase the strength of the interaction with Pol3p. This mutant reverses the genotoxic sensitivity (and cold sensitivity) of the pol32-deletion mutant. This is consistent with the role of Pol32 in stabilizing the polymerase delta complex. The fact that this same mutant does not rescue the defect of pol32-deletion in a BIR assay strongly suggests additional roles for Pol32p (as suggested by other studies), but these are not elucidated. In addition there are a couple of interesting observations such as the fact that destabilizing polymerase delta complex impacts on the retention of Pol1 and Pol3 at HU arrested replication forks. Again these are not mechanistically followed up.

Comments.

The terminology EF463 is unclear. When first introduced (page 5 results) it should be explained.

Changed to E463F464

On page 6. The two mutants do not formally define the conserved domain, they do allow its function to be probed.

Changed to "map to a domain"

Page 9. Why is it interesting that the sensitivity is more pronounced when cells are exposed to both MMS and HU?

Now explained in the text, see Nagai et al., Science 2008

Page 9. Does the increased levels of Pol1 and Pol2 in the pol32-deletion simply reflect an increased time in S phase, where the genes are transcribed?

We mention this in the text. The cells are synchronized (G1 block and released 30 min) so both WT and pol32 cells are entirely in S phase. There is a slightly slower progression of S phase for pol32 delta. Now indicated in the figure that the cells are synchronized in S.

I do not see any reference to figure 5B, which reports that human POLD1 and POLD2 are, as has been reported previously, destabilized upon POLD3 knockdown.

Thank you, this is now referenced.

Referee #2:

In the study, Shimada, et al, identified and characterized novel mutations in POL31, encoding one of the two regulatory subunits of DNA polymerase Pol delta holoenzyme in budding yeast. Specifically, they described pol31-W417 mutation that suppresses cellular sensitivity to MMS, HU and low temperature in pol32 deletion mutant. This is in contrast to the closely related pol31-T415 that seems to phenocopy pol32 deletion. They performed yeast 2-hybrid and in vitro pull-down experiments that suggest pol31-T415 mutation compromises interaction between Pol31 and Pol3 while pol31-W417 improves such interaction in the absence of Pol32. They further presented data that explores the molecular mechanism underlying the phenotypic observations. Importantly they found that the presence of Pol32 as well as Pol31 - Pol3 interaction enhance the stability of Pol delta. This study provides further insight into the function of the Pol31 subunit.

Comments:

1. In the yeast two-hybrid and in vitro pull-down experiments, the authors used two versions of the extreme c-terminal fragments of the Pol3 protein (1097) containing only the CysA + CysB (~100 aa)

or CysB-only (~60 aa) domains. Importantly, the two Pol3 behaved differently. The interaction of CysA+B and Pol31 was not affected by POL32 deletion while CysB - Pol31 interaction was. W417 mutation had no significant impact on CysA+B - Pol31 interaction, while showing dramatically positive influences on CysB - Pol31 interactions. The in vitro pull-down data presented in Figure 3E is less interpretable, as it lacks the WT control at 30 degree. These differences between different constructs raise the concern whether the truncated Pol3 represents the behavior of the full-length protein. Furthermore, the authors have not tested the protein stability of the mutant Pol31, which further complicates the interpretation of the data.

Added 30 degree control for pull down in Supplemental figure S2). Also explained assay better. We explain why the CysA+B binds differently – likely due to larger complex involving PCNA – explained in text.

2. Figure 2E, it is not clear what the authors mean by 16 vs 30 degrees for the pull-down experiments. I suppose it is the temperature at which the cells were cultured before lysed for in vitro pull-down. If this is the case, they have not sufficiently laid out the rationale for such experimental design and discussed the difference observed, since the cell lysate is eventually incubated with a recombinant Pol3-CTD fragment in vitro at 4 degree.

This is now explained in detail and the protocol is described better in the supplemental Figure S2.

3. Figure 3, rev3 Δ showed no sensitivity at the treatment conditions. Is it due to the redundant role of Pol eta in TLS? If this is the case, the data is not sufficient in testing whether the TLS branch contribute to W417 suppression.

The relationship of Rev3 to pol eta is explained in the ms.

4. Figure 3C. The interaction between rad18 and pol32 is a little odd considering both rad5 and mm2 show strong synergy with pol32. The authors may need to double check the strains and maybe repeat the experiment.

The strains are correct and we have explained the observation in the paper. We are afraid that we do not fully understand this comment – see text about role of Rad18.

5. Figure 5C. Since the changes of the expression level of Pol3 is relatively minor, this experiment need repetition and quantitation with error bar.

The experiment has now been repeated 4 times as indicated in the legend and error bars are included

Referee #3:

This is a mutational study of Pol31, the second subunit of DNA polymerase delta (and DNA polymerase zeta). A previous study of Pol31 identified a number of cs/ts mutations and both genetic and physical interactions with several replication/repair pathway proteins, including Mgs1, Rad52, and Srs2. This follow-up study focuses in on two Pol31 mutants. One mutation, T415A, severely compromises interactions with the catalytic subunit Pol3, whereas the other mutation, W417A, shows modestly increased interactions with Pol3, but more importantly for the paper, it partially suppresses the deleterious phenotypes caused by deletion of Pol32, the third subunit of Pol-delta (and zeta). Control studies with rev3-deletion mutants show that the phenotypes are primarily pol-delta related. It is a potentially interesting study, although it may be more suited for a specialized readership than for the DNA repair/replication community in general. Unfortunately, the study is quite uneven in quality, and renewed studies are required to elevate the whole to that of consistent high quality.

Major comments:

1. The data show that the W417 mutation partially suppresses the various phenotypes of the pol32 deletion and moreover that, with the exception of BIR, this suppression is largely pathway independent. The authors suggest that this is the effect of an increased interaction between Pol3 and pol31-W417A. However, it may well be possible that this mutation actually alters the biochemical

properties of the two-subunit complex, such that it shows more robust activity. Wild-type Pol3-Pol31 is compromised for several activities shown by the three subunit enzyme. The Pol31 mutation may enhance some, but not all of them. One interesting finding in this study is that pol31-W417 partially suppresses the PRR, but not the BIR defects of pol32-delta. Does it relate to Pol3-Pol31 levels or its activity?

No, this is not the mechanism because an extra Pol3 copy does not suppress pol32 deletion for BIR but it does for other phenotypes.

This study would be more comprehensive if it were complemented with biochemical studies of the mutant Pol3-31 complex.

This is beyond the scope of the paper

2. Figure 5, Figure S4: It is surprising that authors used C-terminal tagging of Pol3 in several experiments, even while they are cognizant of the fact that this tagging is deleterious for Pol3 function. In contrast, N-terminal tagging (used by others) does not appear to show defects. Without proper tagging, these studies hold little value. But see point 4.

We explain that the N terminal tag was used to attempt ChIP but it is syn lethal with pol32 delta. The Spot-Tag Pol3 complements but is not amenable to ChiP.

3. There is a curious misquoting of the primary literature in this paper. For instance, 4 papers from 2006-2015 are quoted for the monoubiquitination of PCNA by Rad6/18, while the landmark 2001 Nature paper by Hoege et al. is completely missed. Similarly, the identification of CysA and CysB motifs and the placement of the Fe-S cluster has several references, however this analysis originated from a Netz et al. 2011 Nat Chem Biol paper which was not referenced.

Thank you – references are corrected. Sorry for the oversight, it was an inadvertent deletion at last stage of writing.

4. The suppression of pol32-delta by an extra copy of sPot-Pol3 is interesting. However, this altered, tagged form of Pol3 may not be under the same type of regulation as wild-type Pol3. To obtain a clean result, the experiment should be repeated with an extra, integrated, copy of Pol3, containing its proper regulatory sequences.

We have introduced a CEN plasmid (totally stable) and the entire point of that assay is to have elevated Pol3 levels. We have revised the text to clarify this.

Minor comments:

5. Introduction: An inordinate amount of space is taken up in describing the results of the study. The three last paragraphs could be condensed into one small paragraph.

Changed and results are not repeated (condensed into one short paragraph)

6. The compensatory upregulation of Pol1 and Pol2 in pol32-delta is interesting. Can some of this be explained by the increased S-phase residence of the mutant cells?

No, the cells are synchronized in G1 and released synchronously into S and then harvested. They are all in S phase.

June 21, 2021

RE: Life Science Alliance Manuscript #LSA-2021-01138-T

Prof. Susan M. Gasser
Friedrich Miescher Institut for Biomedical Research
Mechanisms of Cancer
Maulbeerstrasse 66
Basel, Basel st. CH-4058
Switzerland

Dear Dr. Gasser,

Thank you for submitting your revised manuscript entitled "The stabilized Pol31-Pol3 interface counteracts Pol32 ablation with differential effects on repair". We would be happy to publish your paper in Life Science Alliance pending final revisions necessary to meet our formatting guidelines.

- Please be sure to upload the final version of the manuscript that includes the responses outlined in your Response to Reviewers
- please upload your main and supplementary figures as single files
- please upload your main manuscript text as an editable doc file
- please add Keywords, Category and a Summary Blurb/Alternate Abstract for your manuscript in our system
- please consult our manuscript preparation guidelines <https://www.life-science-alliance.org/manuscript-prep> and make sure your manuscript sections are in the correct order and labeled correctly
- please add your main, supplementary figure, and table legends to the main manuscript text after the references section;
- please upload your Tables in editable .doc or excel format
- please add an Author Contributions section to your main manuscript text
- please add molecular weights next to all blots

A. FINAL FILES:

B. MANUSCRIPT ORGANIZATION AND FORMATTING:

Sincerely,

Eric Sawey, PhD
Executive Editor

June 24, 2021

RE: Life Science Alliance Manuscript #LSA-2021-01138-TR

Prof. Susan M. Gasser
Friedrich Miescher Institut for Biomedical Research
Mechanisms of cancer
Maulbeerstrasse 66
Basel, Basel st. CH-4058
Switzerland

Dear Dr. Gasser,

Thank you for submitting your Research Article entitled "The stabilized Pol31-Pol3 interface counteracts Pol32 ablation with differential effects on repair". It is a pleasure to let you know that your manuscript is now accepted for publication in Life Science Alliance. Congratulations on this interesting work.

DISTRIBUTION OF MATERIALS:

Again, congratulations on a very nice paper. I hope you found the review process to be constructive and are pleased with how the manuscript was handled editorially. We look forward to future exciting submissions from your lab.

Sincerely,
